



# Technical Note: A new approach to discriminate different black carbon sources by utilising fullerenes and metals in Positive Matrix Factorisation analysis of High-Resolution Soot Particle Aerosol Mass Spectrometer data

Zainab Bibi[1]*, Hugh Coe[1], James Brooks[1], Paul I. Williams[1,2], Ernesto Reyes-Villegas[1], Michael Priestley[3], Carl Percival[4], James D. Allan[1, 2]*

[1]Department of Earth and Environment Science, The University of Manchester, Manchester, M13 9PL, UK.
[2]National Centre for Atmospheric Science, The University of Manchester, Manchester, M13 9PL, UK.
[3]Department of Chemistry and Molecular Biology, University of Gothenburg, Sweden.
[4]Jet Propulsion Laboratory, California Institute of Technology, USA.

Correspondence to: zainab.bibi@manchester.ac.uk, james.allan@manchester.ac.uk

**Abstract.** Atmospheric aerosol particles are known to have detrimental effects on human health and climate. Black carbon is an important constituent of atmospheric aerosol particulate matter (PM), emitted from the incomplete combustion process and cause significant effects on the air quality and human health. Source apportionment of BC is very important, to identify the fraction of BC that has anthropogenic origin and to evaluate the influence of different sources. The High-Resolution Soot Particle Aerosol Mass Spectrometer (HR-SP-AMS) instrument uses a laser vaporizer, which allows the real time detection and characterization of refractory BC and its internally mixed particles such as metals, coating species and rBC subcomponent in the form of fullerene. In this case study, the soot data was collected by using HR-SP-AMS during Guy Fawkes Night on 5th of November 2014. Positive matrix factorization (PMF) was applied in order to positively discriminate between different wood burning sources for the first time, in this case BC from domestic wood burning and bonfires, which no existing black carbon source apportionment technique is currently able to do. Along with this, the use of the fullerene signals in differentiating between soot sources and the use of metals as a tracer for fireworks has also been investigated, which has not significantly contributed to the BC concentrations. The addition of fullerenes signals and successful application of PMF on SP-AMS data apportioned BC into more than two sources. These bonfire sources are hydrocarbon-like Fullerenes, biomass burning organic aerosol, HULIS (humic-like substance) and non-bonfire sources such as hydrocarbon-like OA and domestic burning. The result of correlation analysis between HR-SP-AMS data with previously published Aethalometer, MAAP and CIMS data provides an effective way of quickly gaining insights in relations between the variables and also provide a quantitative estimate of the source contributions to the BC budget during this period. This research study is an important demonstration of using HR-SP-AMS for the purpose of BC source apportionment.


# 1 Introduction

Aerosol particles in the atmosphere are known to have very harmful effects on the air quality, human health and climate
(Highwood and Kinnersley, 2006). One of the most important components of atmospheric aerosol particles is Black Carbon
(BC) i.e. soot, which has extremely detrimental impacts on the human health and air quality (Janssen and Joint, 2012). The
main emission source of BC is through the incomplete combustion of fossil fuel and biomass, involving transportation, open
biomass burning, power generation sources and residential heating sources (Bond et al., 2011; Cooke et al., 1999; US EPA,
2012). In the atmosphere, BC can be mixed with organic and inorganic aerosol species, either at the point of emission or gas-
to-particle conversion processes in the atmosphere.

 As well as harmful impacts on human health, BC can also absorb cancer-inducing pollutants such as volatile organic
compounds (VOCs) and polycyclic aromatic hydrocarbons (PAHs) due to its carbonaceous nature and large surface area. As
a result of its smaller size, can be deposited in weasands and lungs leading to severe health problems (Cao et al., 2012; Dachs
and Eisenreich, 2000). According to hypothesised mechanisms, the ultrafine BC is the cause of abnormal cardiovascular
functions and endothelial senescence at the molecular level (Buchner et al, 2013). Along with being harmful on human health,
it also affects the visibility, reduces agricultural productivity, harms ecosystems and exacerbates global warming (Grahame
and Schlesinger, 2010).

Most BC sources are of anthropogenic origin, but source apportionment is important to establish which specific sources are
responsible. There are multiple measurement techniques available for this purpose but are subject to considerable uncertainties
(Martinsson et al, 2014). Different techniques have been used for the source apportionment of BC. One of the most widely
used techniques is the multiwavelength Aethalometer, which was first described by Hansen et al, 1984. Later Sandradewi et
al (2008) described how Aethalometer can be used to apportion different sources of light absorbing aerosols such as wood
burning, which in contrast to traffic emissions, absorbs additional light in the UV region, over what would be expected in the
near infrared region. Another source apportionment method is to measure the radiocarbon (14C) content (Hellborg et al., 2003).
This method has not been used widely because it requires very specialist equipment (Barescut et al., 2005).

Positive Matrix Factorisation can in principle identify multiple categories of soot, however it needs a large data set and relevant
chemical data of several species. Advancements in different measurement techniques has been deployed by the addition of
Soot Particle Aerosol Mass Spectrometer (SP-AMS) for the online ambient measurements of refractory black carbon (rBC).
In general, the SP-AMS can be operated with the both laser and tungsten vaporiser or with only laser vaporiser. In this study,
the SP-AMS was only operated with the laser vaporiser only in combination with the electron impact source and measures
both refractory and nonrefractory components along with metal nanoparticles (Onasch et al., 2012).

The aim of the current study is to develop the source apportionment tool, which will subsequently improve our understanding
of sources of atmospheric soot. For this purpose, Bonfire night 2014 in Manchester was taken as a case study because it is
known that there were at least three sources of BC (traffic, domestic wood burning, bonfires and potentially fireworks) and
weather conditions that night favoured the high concentrations of primary emissions. This event has been described in previous



studies (Liu et al, 2017: Priestley et al, 2018; Reyes et al, 2018). In terms of air quality status, it has been recognised that Bonfire night is one of the most polluted days in the UK. Every year, this event is celebrated on 5th of November where open fires are lit and fireworks are set off at individual households, as well as part of the community events. These bonfire activities have a strong flaming segment which roughly start during the evening and lasts for up to 2 hours. The fires after flaming are

not refuelled, therefore leading to an extended phase of smouldering as the fires are left to completely burn and die down (Dyke et al.,1997; Mari et al., 2010; Pongpiachan et al., 2015).

Different research case studies have previously been published about the Bonfire night around UK. For example, Clark (1997) studied the PM10 concentrations emitted during the Bonfire night event in different parts of UK. In Oxford, dioxins measurements in the ambient air were conducted by Dyke et al (1997). Colbeck and Chung (1996) targeted the particle size

distribution. The polycyclic aromatic hydrocarbons (PAHs) were measured in Lancaster (Farrar et al., 2004), while in 2018, Reyes et al, studied about the insights into the nitrate chemistry during the bonfire night by applying the chemical ionisation mass spectrometry measurements and aerosol mass spectrometry simultaneously (Reyes et al., 2018). Observations of the Nitro-compounds including nitrate, amide and isocyanate were studied during the bonfire night in Manchester (Priestly et al., 2018). In previous studies, specifically during the bonfire event and general aerosol measurements, several different source

apportionment techniques have been performed. Aethalometer AE31 model was performed to do the source apportionment analysis and successfully apportion the rBC into BC from wood burning and BC from the traffic emissions (Reyes et. al., 2018). During the same study, Multilinear Engine-2 and PMF tools were also used over the AMS data through the source apportionment interface (SoFi version 4.8 as presented by (Canonaco et al., 2013) in order to find the organic aerosol sources according to proposed strategies by Reyes et al., (2016) and recommendations made by Crippa et al. (2014).

The HR-SP-AMS is a combination of Single Particle Soot Photometer (SP2) and High-Resolution Time-of-Flight Aerosol Mass Spectrometer (HR-ToF-AMS). Based on the design used in SP2, the SPAMS is equipped with an intracavity laser vaporiser Nd: YAG (1064 nm), that heats up and vaporise both core and coating particles, which was collected before, during and after the Bonfire night event in UK, concurrent with the previous measurements. In order to vaporise the refractory particles types that are not detected by the standard AMS, the new vaporiser is designed, to detect the vaporised species through electron

ionisation for the generation of chemical ion, thus keeping the vaporisation and ionisation steps separate. The new intracavity laser vaporiser allows AMS to characterising the refractory chemical components of ambient aerosol species (organics and inorganics), particularly, including light absorbing refractory Black Carbon (rBC) particles (Onasch et al., 2012).

In order to test the ability of HR-SP-AMS to apportion rBC (with multiple BC types) the data was collected during bonfire night from 29th Oct-11th Nov 2014 at the University of Manchester. As a result of strong meteorological conditions, very high

and mixed concentrations of pollutants were observed. Traditionally the PMF tool is applied to conventional AMS data (as with Reyes et al., 2018) but the objective of this study is to demonstrate a new way to source apportion black carbon based on highly time resolved mass spectrometric composition data of the population of particles that contain black carbon and uses information on the composition of black carbon and information on internally mixed fullerenes and condensed material.



Fullerenes are a class of exclusively carbon compounds with polyhedral closed-shell structure. They were identified as ionized particles in low-pressure fuel rich flat premixed acetylene and benzene–oxygen flames by molecular-beam sampling combined with mass spectrometer analysis (Gerhardt, Loffler and Homann, 1988). These have been reported previously in HR-SP-AMS data (Fortner et al., 2012).

## 2 Methodology

### 2.1 Sampling site overview

Measurements were conducted at the South Campus University of Manchester (53.467°N, 2.232°W) before, during and after the bonfire night event on 5th November as described in the previous publications (Liu et al., 2017; Reyes et al., 2018). The sampling station is surrounded by the nine public parks. Different instruments were set up for the online measurement of ambient aerosols and gases. A Compact Time-of-Flight Aerosol Mass Spectrometer (cTOF-AMS) was used to measure all PM1 components as described by Reyes et al., 2018. A Time-of-Flight Chemical Ionisation Mass Spectrometer (ToF-CIMS) was used to measure the gas phase concentrations of aerosols (Priestley et al., 2018). BC source apportionment was performed by using Aethalometer AE31 which measured the absorption of light at seven different wavelengths (Reyes et al., 2018) while a MAAP was used to measure the concentrations of BC emitted during the bonfire event and inform the corrections needed to process the AE31 data (Collaud Cohen et al, 2010).

### 2.2 Instrument Overview:

In this case study, the High-Resolution Soot Particle Aerosol Mass Spectrometer (HR-SP-AMS) used was not the same as the Compact Time of Flight AMS featured in Reyes et al., (2018). A catalytic stripper was also attached to the aerosol sampling lines, which switched between catalytic stripper and direct measurements after every 30 minutes (Liu et al., 2017). In our case, the results have been analysed by using the direct measurements only.

During the experiment, a measurement of the ionisation efficiency was not obtained owing to technical difficulties generating and detecting a suitable test aerosol. Therefore, while the concentrations are not quantitative, relative proportions should still be valid and proportional to absolute concentrations, so this dataset should still serve as a technical proof-of-concept for factorisation. The HR-SP-AMS data were analysed using the data analysis toolkit TOF-AMS HR Analysis 1.20O (DeCarlo et al., 2006). The high-resolution feature of toolkit allow the direct separation of most ions from the organic and inorganic species at the same nominal mass by charge ratio, the separation of BC family from its coating species, the direct identification of organosulfur and organic nitrogen content and also the quantification of many different types of their organic fragments (isotopes) such as $C_xH_y$, $C_xH_yO_z$, $C_xH_yO_zN_p$ (Aiken et al., 2007). This high-resolution analysis on SP-AMS data also detected various metal pollutants such as Iron (Fe), Titanium (Ti), Strontium (Sr) and Caesium (Cs).





The instrument alternated between three mass spectrometer configurations: The standard 'V' mode, high-resolution 'W' mode (De Carlo et al., 2006) and an alternative 'V' mode whereby the orthogonal extractor was pulsed every 95 instead of 34 μs.
This lower frequency delivered data up to m/z=3200 rather than 380, with the intention of characterising the fullerene signals described by Onasch et al. (2012) at the expense of overall signal-to-noise. The data presented in this paper are a combination of the standard 'V' mode for the lower m/z peaks, processed using the PIKA high resolution analysis tool, and the long pulser period 'V' mode for the fullerene peaks, processed using unit mass resolution (UMR) method. The 'W' mode data was deemed not sensitive enough to contribute to this work.

## 2.3 Positive Matrix Factorisation

Positive Matrix Factorization (PMF) is an advanced factor-analysis technique developed by Paatero and Tapper (1994). In the previous researches, PMF has been used extensively to apportion organics with the standard AMS data but not so often to apportion BC from SP-AMS data (Crippa et al., 2012). In this research study Positive Matrix Factorization (PMF) was applied on HR-SP-AMS data to apportion BC in to more than two sources. PMF assumes that a matrix of data can be explained by a
linear combination of "factors" with characteristic profiles and varying temporal contributions (Paatero and Tapper, 1994; Ulbrich et al., 2009). The analysis was conducted using the PMF Evaluation Tool (Ulbrich et al., 2009; Zhang et al., 2011). Because of the lower signals and the different data retrieval method used for the fullerene signals (UMR rather than HR), greater emphasis had to be placed on these signals. This was done by applying an additional 'model error' to the error matrix, i.e. an error term proportional to the signal intensity instead of its square root, as per the standard AMS error model (Ulbrich
et al., 2009). As well as placing greater emphasis on the smaller fullerene signals, the application of this model error also increased the number of "weak" variables, defined as having SNR below 2 (Paatero and Hopke, 2003; Ulbrich et al., 2009), which were down weighted by a factor of two. No variables were "bad" in the sense of having SNR < 0.2 (Paatero and Hopke, 2003). Additional details and residual plots are available in the supplementary material.

## 3 Results

### 3.1 Weather measurements and overview of highly polluted time period:

The weather data is as presented by Reyes et al, (2018), and results showed quiet stagnant conditions with the low temperature of 4ºC, high relative humidity of 85% alongside the wind speed of 1.5 m/s and varying wind directions. This type of weather condition facilitates the increasing amount of pollution in the atmosphere. During the periodic stagnation weather phenomenon, the very high concentrations of BC was also observed with the signal of 3400/s during the bonfire event at 10:20 pm as
compared to (100 s$^{-1}$ – 500 s$^{-1}$ before Bonfire and 250 s$^{-1}$ - 300 s$^{-1}$ after bonfire night) fig 1. The time period of the bonfire night when the pollutants were very high is called as high pollutant concentration time period.



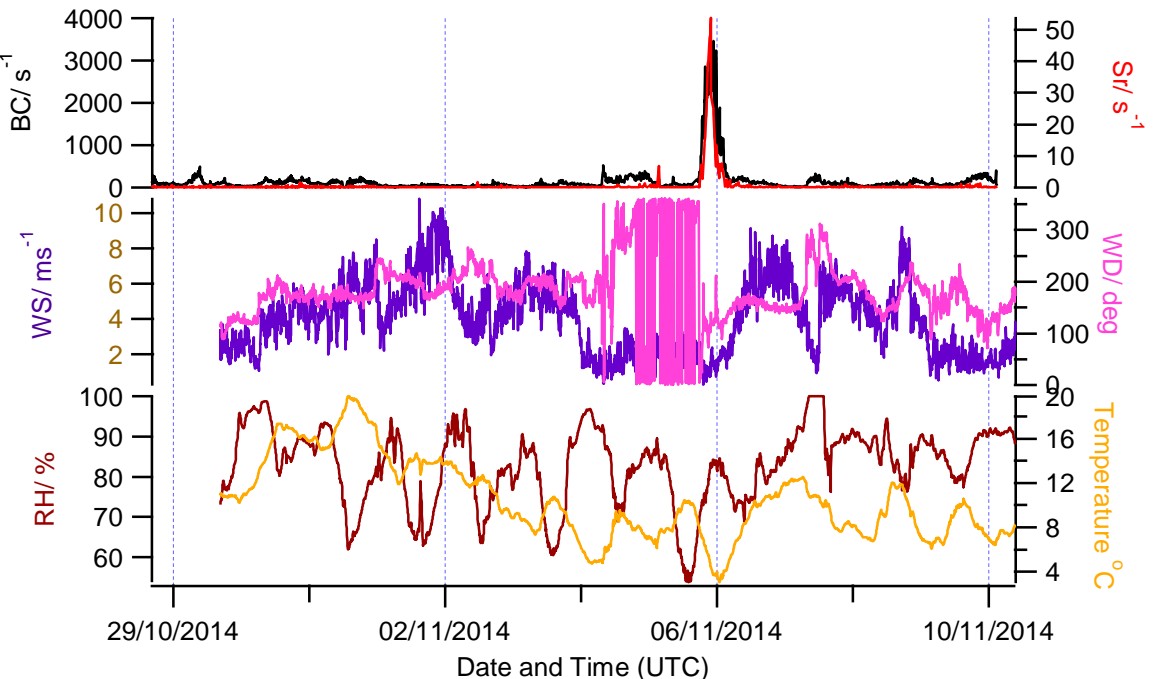

**Figure 1: The meteorological measurement (Relative humidity, Temperature, Wind direction and Wind speed) along with the time series of BC and firework tracer emitted during the bonfire night.**

### 3.2 Firework burning tracer:

In order to identify a unique tracer for fireworks, the HR-SP-AMS data was analysed for metals. Reyes et al., (2018), concluded that fireworks were not a major factor but could not directly support this assertion with the data available. However, HR-SP-AMS offers insight. Fireworks release several pollutants such as manganese, cadmium, strontium, aluminium, and other suspended particles, carbon monoxide, carbon dioxide and sulphur dioxide etc. (Lemieux et al., 2004; Shi et al., 2011). These metal compounds are in the form of metal salts such as potassium chlorates, perchlorates, strontium nitrates, potassium nitrates, barium nitrates, charcoal, sodium oxalate, manganese, sulphur, iron, aluminium etc. Mainly these metals can used to give different bright colours, for example Sr can be used for giving red colour to the fireworks (Mclain, 1980). And during the analysis, different metal peaks such as Iron (Fe), Strontium (Sr), Caesium (Cs) and Titanium (Ti), that could be associated with the fireworks were detected (fig 2). The Sr was most unambiguously associated with the fireworks due to the fact that there is no other signal present in the atmosphere outside of Bonfire night. Other metals may have other sources, such as mineral or brake dust in the case of iron or may be receiving signal interference from other mass spectral peaks. The highest peak of Sr concentrations i.e. 53.6 $s^{-1}$ was detected as compared to the concentrations of Sr, before and after Bonfire event (1.6 $s^{-1}$ and 0.9 $s^{-1}$).



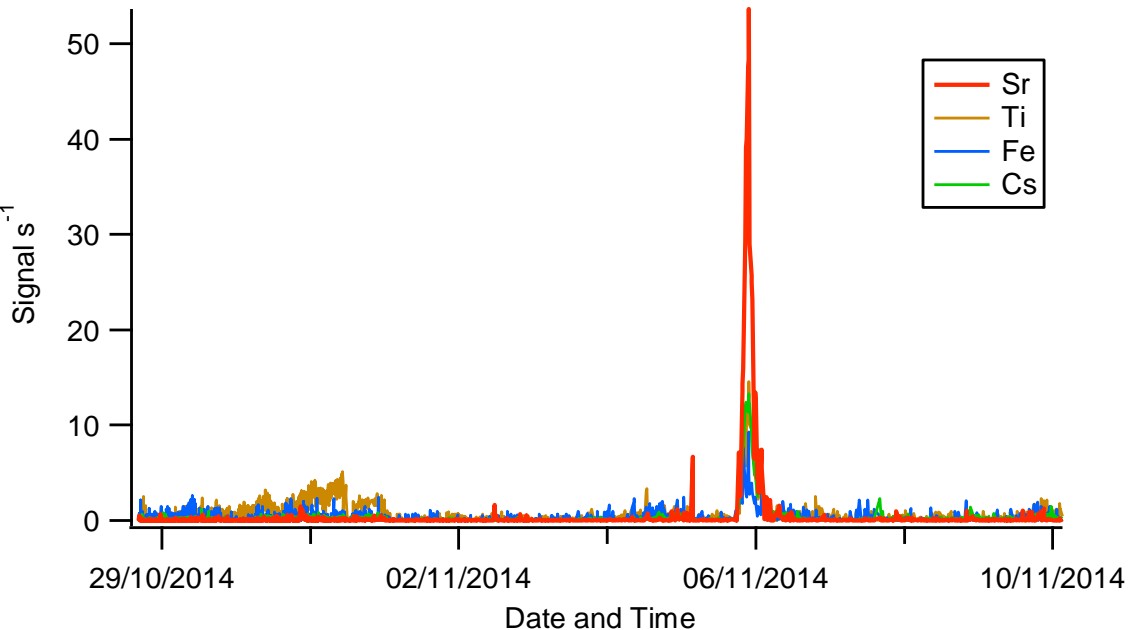

**Figure 2: Time series of various metal pollutant concentrations observed during the bonfire event.**

### 3.3 HR Time series of BC and its coating species

Figure 3 shows that the signals associated with refractory BC (rBC) and its coating species (Org, $SO_4$, $NO_3$, Chl and $NH_4$) were particularly very high during the bonfire night. The HR-time-series of the whole sampling time-period shows that the majority of non-refractory $PM_{BC}$ signal was mainly organic matter, having very high concentrations, followed by Chl, $SO_4$,

$NH_4$ and $NO_3$. It has been worth mentioning that the signals of these aerosols were very high during the bonfire night as compared to before and after time period except the $NO_3$ signal which was 0.8 $s^{-1}$ before bonfire event on 30th Oct at 8:30 am and 1.8 $s^{-1}$ after bonfire night on 7th Nov at 4:15 pm.




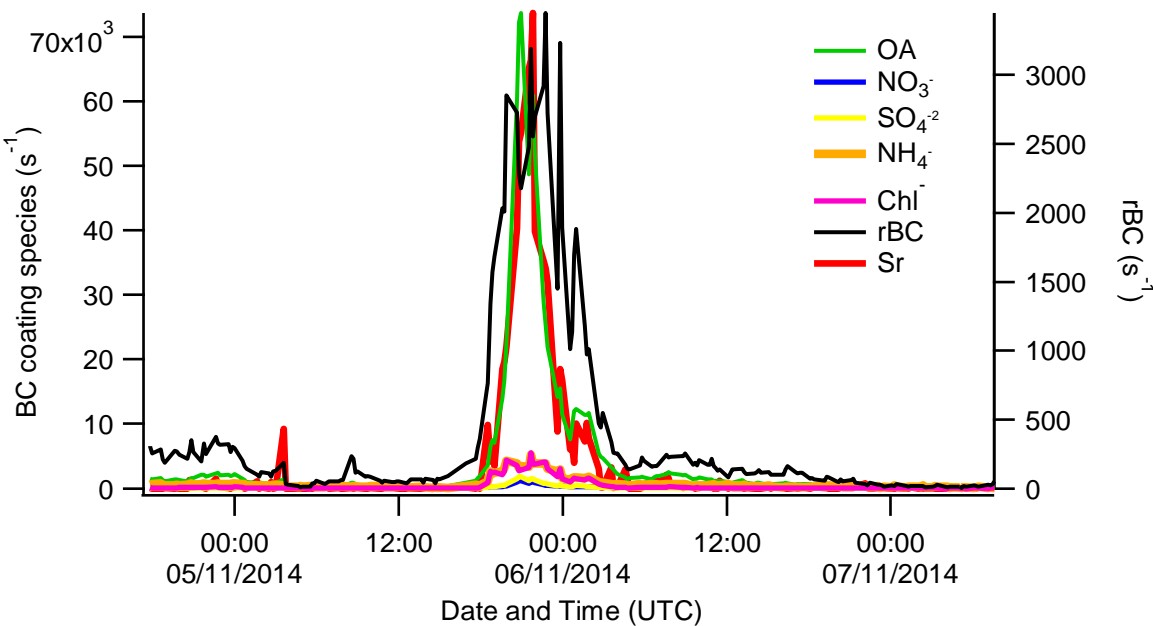

**Figure 3: Time series of High Resolution rBC concentrations and its coating species (Organics and Inorganics) measured during the all measurement time period.**

The high Chl peak was strongly related with the open fire burning that happened on the Bonfire night. Wood burning is an important source of chloride in the atmosphere (Lobert et al, 1999). Fireworks can also be a likely source of Chl, because chlorates and perchlorates can provide oxygen for the combustion of fireworks. Also, the high peak of nitrate can be linked with the combustion sources such as wood burning and biomass burning emissions (Reyes et al. 2018). The concentrations of organic aerosol started increasing at 8:30 pm to 9:00 pm (highest), while concentrations of rBC were increasing after 1:50 hour later.

### 3.4 Correlation analysis of rBC with other pollutants

The HR-SP-AMS data was compared against those of other instruments such as AE31, CIMS, MAAP and AMS presented in the previous studies (Reyes et al, 2018; Priestley et al., 2018) and a statistically significant correlation was found between the BC measured by three different instruments i.e. rBC from HR-SP-AMS, BC from MAAP and BC from AE31 (fig 4). A very high concentration of rBC and BC was measured from all instruments that could detect these. The peak of Brown Carbon (BrC) measured by AE31was also very high during the event night and indicates a wood burning source (details found at Reyes et al., 2018).





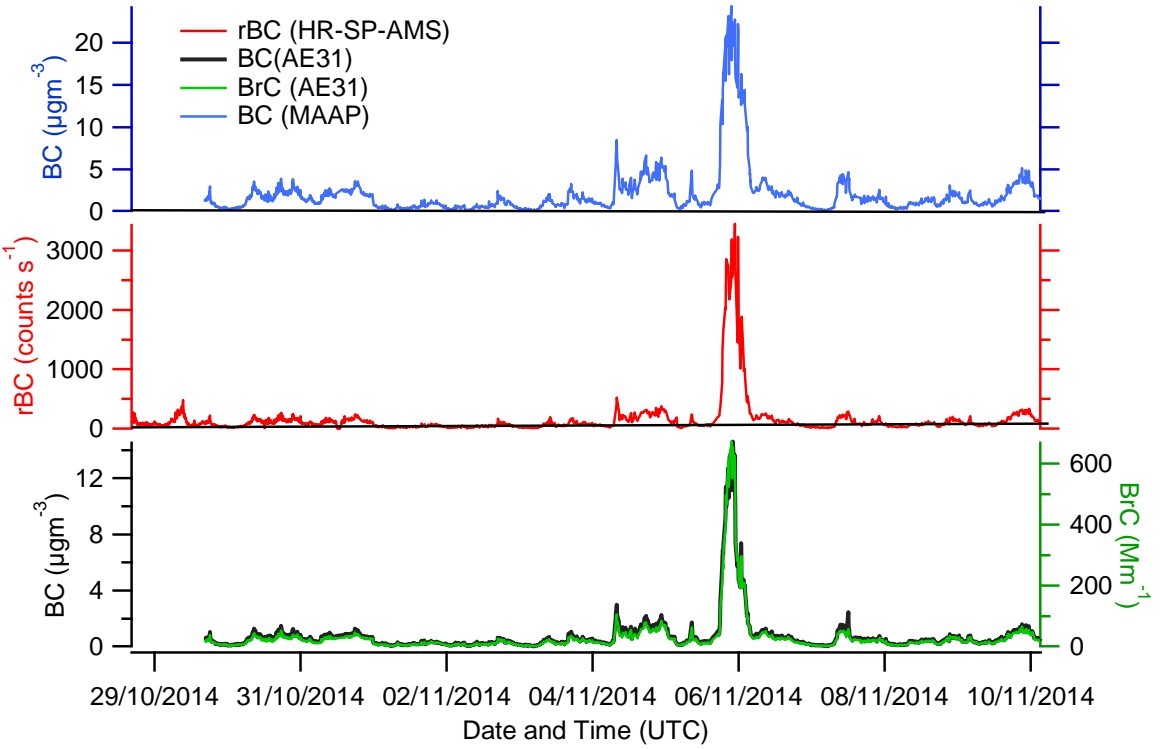

**Figure 4: Time series of Black Carbon measured by different instruments (HR SP-AMS, AE31 (BC and BrC) and MAAP) during Bonfire night event.**

Reyes et al., (2018) used AMS to estimate the concentrations of Particulate Organic Oxides of Nitrogen (PONs) i.e. 2.8 $\mu$gm$^{-3}$ by using m/z 46:30 ratios according to previous literature. The study also identified two PON factors into primary PON and secondary PON by applying ME-2 source apportionment on organic aerosol concentrations from different sources after modification of fragmentation table. Figure 5 shows the time-series of rBC, Primary Particulate organic oxide of nitrogen (pPON) and Secondary Particulate organic oxide of nitrogen (sPON). The result show that the concentration of up to 2.8 $\mu$gm$^{-3}$ for PON was detected which was over the detection range as reported by Bruns et al. (2010). Moreover, BC was also detected with very high signals of 3400 s$^{-1}$.





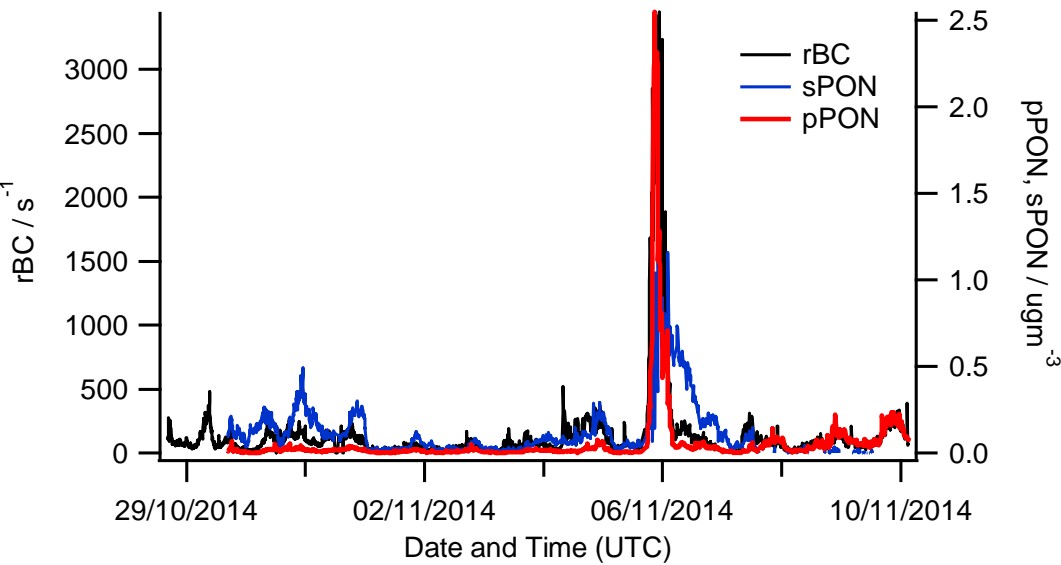

**Figure 5: Time series of rBC, primary (pPON) and secondary (sPON) organic nitrate measured during the biomass burning event.**

The reason behind this high correlation is that, rBC is a primary pollutant, so it is well correlated with the primary PON as both are directly emitted from the bonfire event. While the correlation of rBC with the sPON is not very good i.e. $r^2 = 0.35$, because the secondary pollutants appeared later.

### 3.5 BC Source apportionment analysis

#### 3.5.1 PMF factor profile

The first stage of performing PMF analysis is to select the optimum number of factors. A stepwise approach was used to select the number of factors for PMF, beginning with a 2-factor model and successively adding factors up to a maximum of six. In our case, five-factors gave the best solution based on the criteria of $Q/Q_{exp}$ near 1, the squares of scaled residuals' total sum and all the matrix points fitted within their expected error (Paatero et al., 2002). The rotational ambiguity of the five-factors solution was explored by varying the f-peak between -2.0 and +2.0 with fpeak interval of 0.2. Along with that the changes in the fractional contribution of the PMF factors was very small for all the factors, indicating that changing f-peak value over a range of 2 and -2, (away from 0) did not affect the overall results of PMF analysis. The solution for fpeak=0 was used for all subsequent work, as also recommended by Paatero et al. (2002). Adjustments to the model error value were also made in this analysis because, in addition to the fullerene issue discussed previously, the true factors are not constant as assumed by the PMF model (Comero et al, 2009). The model error value was increased to 0.10 in order to reflect that the data and uncertainties have a lognormal distribution and to upweight the fullerenes signals. (see supplement).





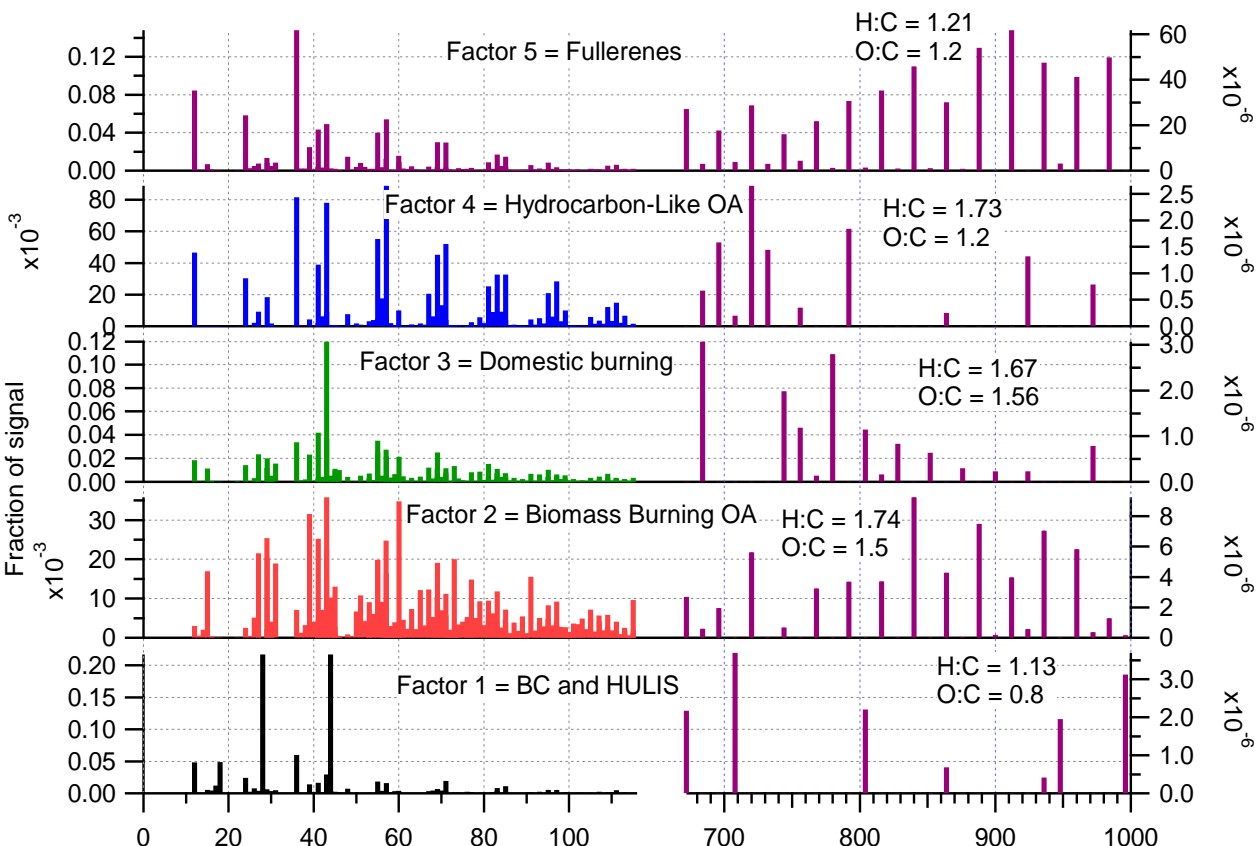

**Figure 6: PMF five factors source profile (factor 1 = BC and HULIS, factor 2 = BBOA, factor 3 = Domestic burning**
**OA, factor 4 = Hydrocarbon-Like OA, factor 5 = Fullerenes). Note the difference in scales of the fullerene signals (right**
**hand axes).**

Fig 6 is showing the profile concentrations and factor contribution's mass spectra of five different sources determined by PMF analysis. Three of them (BC and HULIS, biomass burning OA and Fullerenes) are directly linked with the bonfire night while HULIS and other two (domestic burning and traffic emissions) are the additional sources and are called as non-bonfire night sources. HULIS is a class of organic molecules that seems to be formed secondarily by photochemical oxidation, deoxidization, and polymerization of volatile organic compounds in the atmosphere (Hoffer et al., 2004) and biomass burning (Lin et al., 2010), with a characteristic peak at m/z 44 (Kiss et al., 2003). Potential origins of HULIS in the atmosphere are diverse, including primary terrestrial (Simoneit, 1980) and marine sources (Cini et al., 1994; Cini et al., 1996; Calace et al., 2001; Cavalli et al., 2004), biomass burning (Graber and Rudich, 2006; McFiggans et al, 2005; Mukai and Ambe, 1986; Zappoli et al., 1999; Graham et al., 2002; Mayol-Bracero et al., 2002), and secondary organic aerosol formation (condensation, reaction, oligomerization, etc.) (Gelencser et al., 2002; Jang et al., 2002; Jang et al., 2003; Tolocka et al., 2004; Hung et al., 2005). Moreover, the fullerene signals were added to detect more than two sources of soot by applying PMF on HR-SP-AMS data.



Although it is not clear why fullerenes are sometimes observed, it does seem to differentiate between biomass burning during the bonfire event and biomass burning from domestic burning. In fig 6, the Factor 5 was heavily weighted by hydrocarbon like

fullerenes having a peak at m/z 720 ($C_{60}^+$), which was typically not associated from the traffic source (diesel) but came from the emissions changes depending on the different phase of combustion during that time-period of bonfire event. Factor 4 resembled Hydrocarbon like organic aerosols (HOA) and is related to the traffic emissions (fossil fuel combustion), presenting the high signals at m/z 55 and m/z 57 typically aliphatic hydrocarbons (Canagaratna et al., 2004). Diesel exhaust is typically dominated by re-condensed engine lubricating oil and consists mainly of n-alkanes, branched alkanes, cycloalkanes, and

aromatics (Canagaratna et al., 2004; Chirico et al., 2010), leading to high signal at the ion series $C_nH_{2n+1}^+$ and $C_nH_{2n-1}^+$. In particular, m/z 57 is a major mass fragment and often used as a tracer for HOA (Zhang et al., 2005).

Factor 3 presented a relatively mixed factor source having the high loading of signals at m/z 43, m/z 55, m/z 57 and m/z 60. Therefore, this factor was mainly related to the secondary domestic wood burning sources because of its high peaks observed before and after the event night. Factor 2 was specifically loaded at m/z 60 and m/z 73 (levoglucosan), indicating the typical

source profile of by wood burning organic aerosols (Alfarra et al, 2007). In previous AMS studies, cooking could be one of the important sources of $PM_{2.5}$ (Sun et al., 2013), but in this study cooking was not identified by PMF because it is not co-emitted with the rBC so is not vaporised by the SP-AMS. The factor 1 having signals at m/z 36 related to BC (the main bonfire emission source) and m/z 44 is highly related to HULIS, showing its contributions from bonfire night as well as secondary sources.

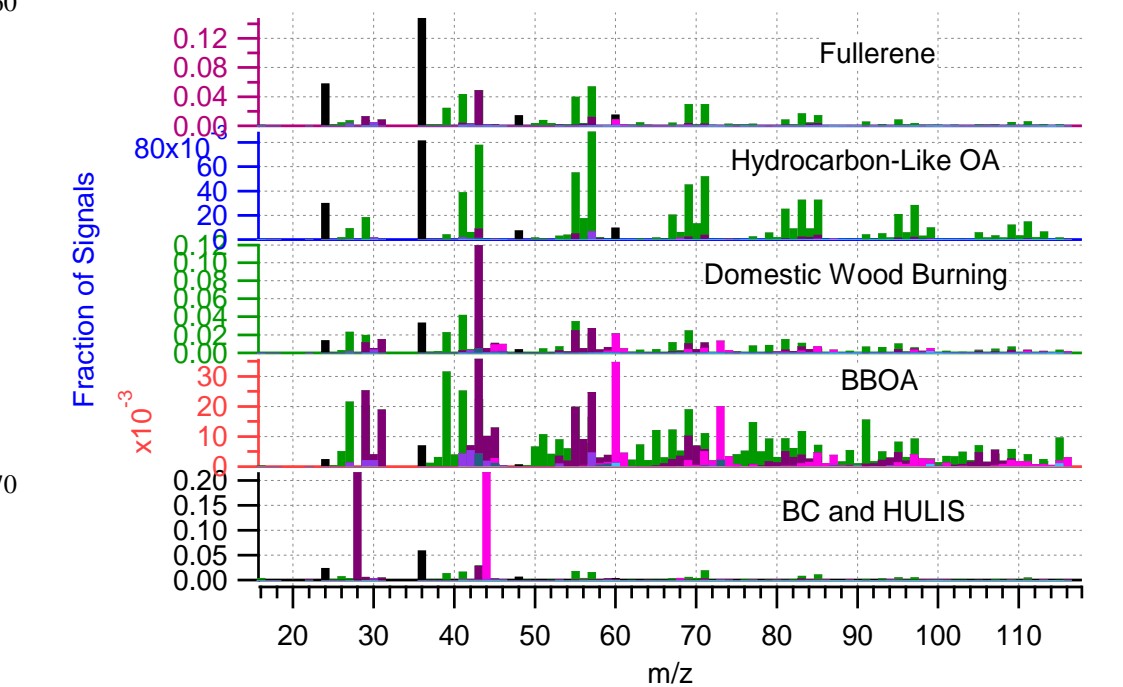

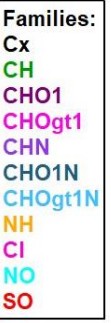





**Figure 7: The family coloured spectra of 5 factors solution during the BF event.**

Fig 7 showed the HR mass spectral profiles in which ions with different oxygen and hydrogen contents (different ion families) are highlighted by the different colours (Aiken et al, 2009; Aiken et al, 2010). Each factor profile has distinct, unique masses between m/z 12 to m/z 116 range. In factor 1, family CHOgt1 (CHO fragments with more than one oxygen atom) with the high peak of m/z 44 represent secondary aerosol, HULIS and another high peak at m/z 36 representing BC. The factor 2 is mainly dominated by the CHOgt1 at m/z 60 and m/z 71 and is biomass burning factor. While factor 3 is mixed factor having peaks from different families such as CHO1 (CHO fragments with only one oxygen atom), CH (CH fragment), CHOgt1 (CHO fragments with more than one oxygen atom) and BC which is depicting both primary and secondary aerosols and hence this factor present the domestic burning sources. In factor 4, mainly CH family is dominated which is pointing towards the emissions of HOA mainly from the diesel traffic emissions and BC (C3+) emissions from traffic. Factor 1 is dominated by primary emissions i.e. fullerenes with some BC.

### 3.5.2 PMF time series

Fig 8 shows the separately plotted Bonfire night factors and non-bonfire night factors. Factor 1 (BC and HULIS), Factor 5 (HOA-like Fullerenes) and factor 2 (Biomass Burning OA) are the bonfire night factors having very high peaks detected only during the bonfire night while factor 4 (Hydrocarbon-Like OA) and factor 3 (domestic wood burning), are the two non-bonfire night factors, which are behaving completely independently, as their peaks have been observed before, during and after the bonfire night. The domestic burning source can be related to activities such as household wood combustion. In order to test whether any of the factors could be associated with fireworks, PMF analysis was also performed to force the inclusion of Sr in the factorisation. For this purpose, the Sr concentrations were upweighted by multiplying the total concentrations of Sr (but not the associated error) by 10, 100 and 1000, but despite this, a factor containing Sr was not found. This implies that none of the HR-SP-AMS factors could be associated with fireworks.



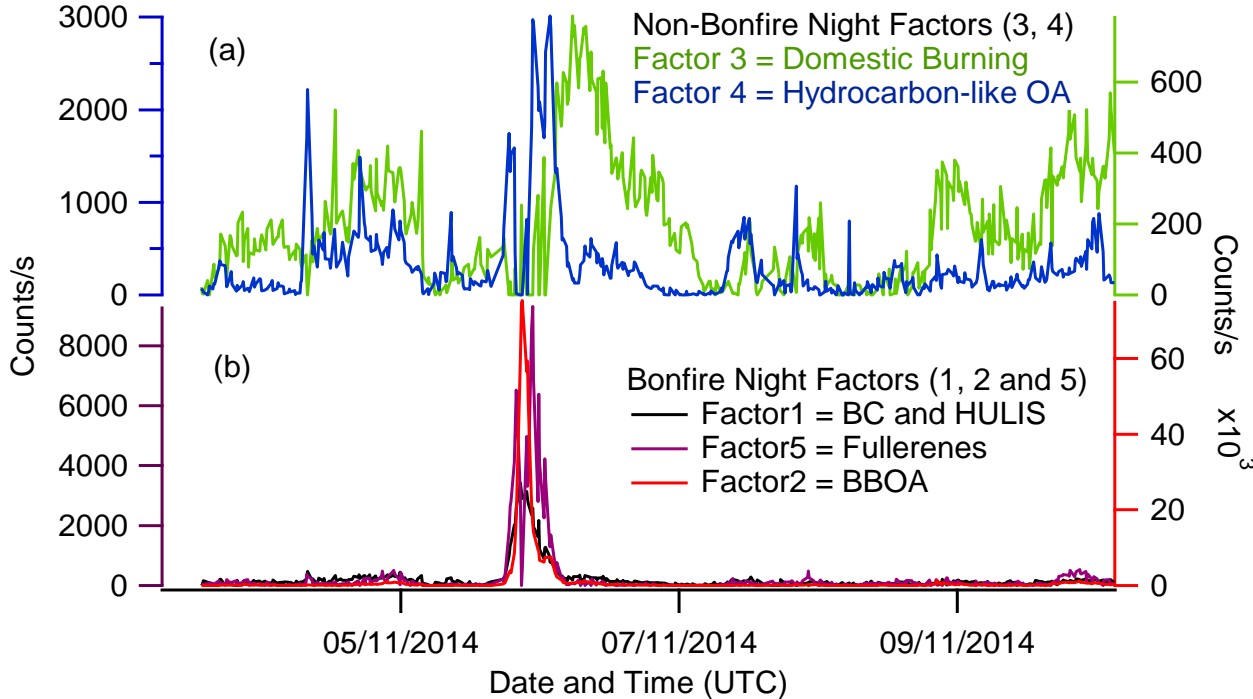

**Figure 8: The time series of (a) non-Bonfire night factors and (b) bonfire night factors obtained by PMF analysis.**

## 4 Discussion

### 4.1 Correlation between different pollutants

The correlation plot gives an effective way of quickly gaining an idea of how variables are related with one another. The data analysis software 'openair' was used to generate the hierarchical cluster analysis chart (Carslaw and Ropkins, 2012) using the 'corplot' function on the bonfire night data only. Hierarchical cluster analysis will provide an effective way of understanding of the order how different variables appear due to their similarity in one another. A dendrogram was plotted to provide an additional information to help visualising how groups of variables are related to one another. The explanation of all these time

series names and how do they measured is given in table 1 in the supplementary material.





**Figure 9: The similarity between different pollutants with one another, through hierarchical cluster analysis (HCA)**

In fig 9, a significant correlation was observed between Fullerenes, BC$_{total}$, HONO, HCN, rBC. The reason is that all of these are primary pollutants and directly released from the bonfire emissions. HCN and HONO are the nitrogen containing gases that has been released during the bonfire night from the wood fires (Le Breton et al., 2013; Wang et al., 2015). BC$_{tr}$ has also shown the strong correlation with the rBC because it is also the source of primary pollutants and have direct influence from all the bonfire factors. The HULIS and BBOA has very close relationship which depicts that both are bonfire factors but on the other hand HULIS also have some relationship with the secondary sources, so it also has some contributions towards the non-bonfire factors. Hydrocarbon-like OA and sPON has close similarity due to the reason that just like sPON, the peak of Hydrocarbon-like OA has been observed before and after the event night. Sr and pPON and HCNO are showing similarity but 'Sr' is behaving as a complete separate factor. While pPON and HCNO both are primary pollutants. The reason for their correlation with Sr is due to this fact that their peaks are occurring at the same time period but from different sources. However,





an obvious correlation was seen between BrC and BC$_{wb}$, because BrC is mainly released due to the emissions of wood burning

product. And the last factor i.e. domestic burning is behaving as a separate factor and showing no or very less correlation with

any sources.

Based on HCA plot, a time series graph was also plotted to investigate the timings of all pollutants having strong relationship

among one another (fig 10). The second group which showed the strong correlation was HCNO and factor 5 (hydrocarbon-

like fullerene) with the $r^2$ value of 0.96. While Isocyanic acid (HNCO) is another highly toxic, long-lived gas (lifetime of days

to decades; (Borduas et al., 2016) emitted from biomass burning (BB) with similar anthropogenic and biogenic sources as

HCN. Urban sources of HNCO are attributed to primary activity such as automotive emission (Jathar et al., 2017), residential

heating (BB) (Woodward-Massey et al., 2014), and industrial processes (Sarkar et al., 2016).  Specific reason for their very

close relationship is due to their same emission sources but at different temperatures during the bonfire night.


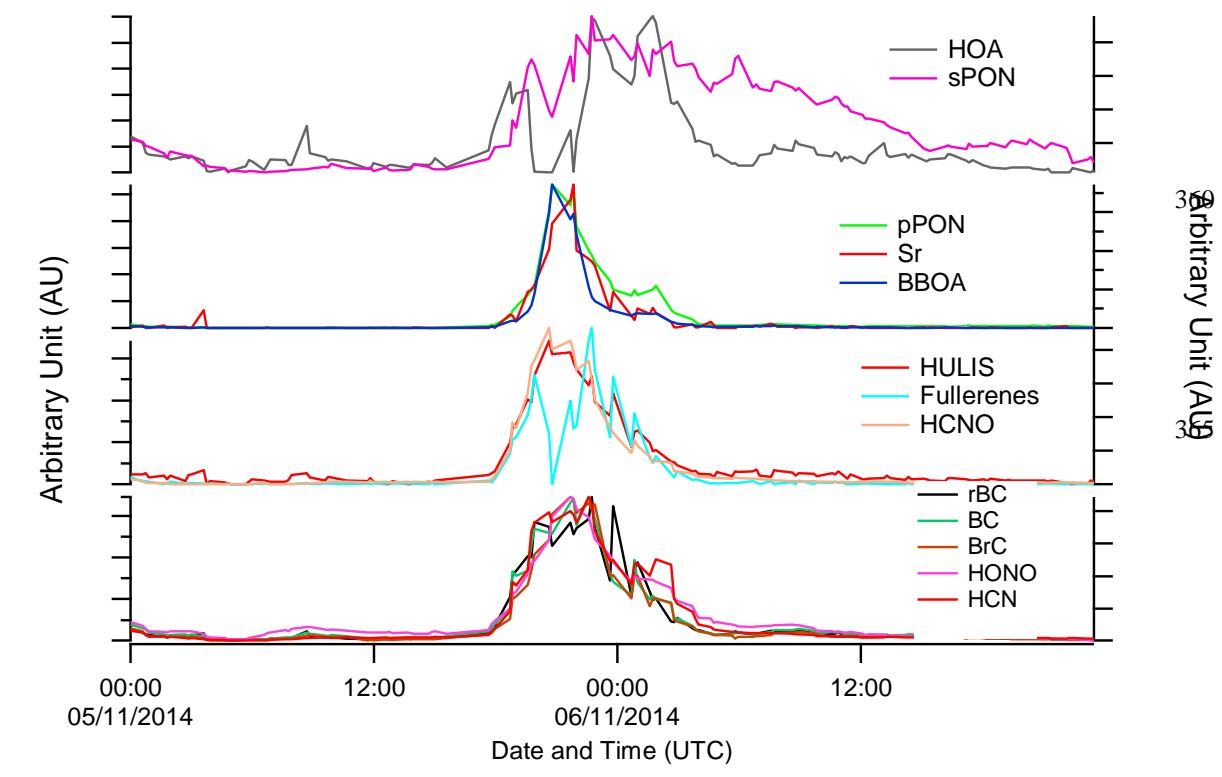

**Figure 10: Time series of the pollutants, grouped according to the Hierarchical Cluster Analysis in figure 9.**

Another close correlation was factor 1 (HULIS) with an $r^2$ of 0.82 with fullerenes. The reason for this high correlation is

because of the very high concentration released during the bonfire event. In terms of sPON and factor 4 (traffic emissions),

both are showing moderate correlation i.e. $r^2$= 0.64. The $r^2$ in this case is much lower than the other grouping because traffic





emissions are the primary source, not a secondary source, so their relationship is likely coincidental, maybe modulated by the boundary layer. The last group of pollutants having close correlation included pPON, Sr and factor 2 (biomass burning OA).

In a previous research study (Reyes el al., 2018), ME-2 analysis indicated the presence of two different types of PON, in which particularly pPON are primarily emitted along with BBOA concentrations. According to Zhang et al., (2016), PON are related mainly to the primary sources of combustion.

The firework tracer Sr has shown some correlation with pPON and BBOA, but their peaks occurred at slightly different times. So, in spite, of the high correlation, this implies that they're not identical. It could be that if the firework display occurred at

the beginning of the bonfire event their emissions maybe coincident with the pyrolysis emissions on bonfire begin to be lit, as distinct from the smouldering emissions later (Haslett et al., 2017), however, without specific knowledge of the timings of the events that contribute to these emissions, it is difficult to conclude this. Coupled with the fact that Sr could not be incorporated with any of the factors in this study, this would be consistent with the notion that fireworks are not the soot sources reported here or in Reyes et al. (2018).

**4.2 Relative contributions of the different factors to the HR-SP-AMS signal and the BC budget:**

This data can be used to estimate the relative contributions of the different sources to the overall signal and the black carbon assuming that the divergence of the aerosol in the beam is the same for all particle types and hence the efficiency is same for all particle types. Fig 11(a) illustrates the total signal fraction of BC accounted by each BC sources released during the Bonfire event. The total mass fraction was obtained directly by the PMF analysis.

The five factors have been divided into two different categories i.e. Bonfire factors and non-Bonfire factors. The Bonfire factors are HOA-like-Fullerene and BBOA, while HOA and domestic burning are the non-bonfire sources and HULIS having contributions from both. The biggest contribution from the event was BBOA, contributing 54% out of total signal fraction followed by traffic emissions (13%), HOA-like-Fullerenes (11%), HULIS (9%) and domestic burning (8%). Fig 11 (b) shows the mass fraction of only black carbon from each PMF factor profile. The

BC only mass fraction was calculated by multiplying the total signal fraction with the fraction of rBC in each factor and then renormalize to 1. According to analysis, HOA-like-Fullerenes contributed the highest fraction i.e. 42% followed by the non-bonfire factor HOA (traffic emissions) with 27% contribution. The HULIS and BBOA have 13% and 10% contribution respectively while domestic burning has the least part with 7% only. Therefore, it has been clearly found that the two major sources of BC are HOA-like-Fullerenes and traffic emissions. While in fig

12 shows the quantitative data of BC signals in µg m$^{-3}$ after scaling them to the AE31 BC$_{950}$ data. This time series was generated by following the same procedure for BC signal fraction out of total signal fraction and normalising to the total BC signal. According to the time series, BC from Fullerenes and BBOA showed the highest signals during the bonfire night-time period followed by HULIS, domestic burning and traffic emissions.






**11a**


Bonfire factors


Non-Bonfire factors

Hydrocarbon-like Fullerene (11%)

Biomass Burning OA (54%)

HULIS (9%)

Hydrocarbon-like OA (13%)

Domestic Burning (8%)

Residual (5%)

Profile Factors

Total signal fraction

**11b**

Bonfire factors

Non-Bonfire factors

Hydrocarbon-Like Fullerene (42%)

Biomass Burning OA (10%)

HULIS (13%)

Hydrocarbon-like OA (27%)

Domestic Burning (7%)

Residual (-1%)

Profile Factors

rBC signal fraction

**Figure 11: Fig 11(a) shows the relative contributions of total mass fraction to HR-SP-AMS signal and Fig 11(b) shows rBC mass fraction accounted for by each PMF factors.**





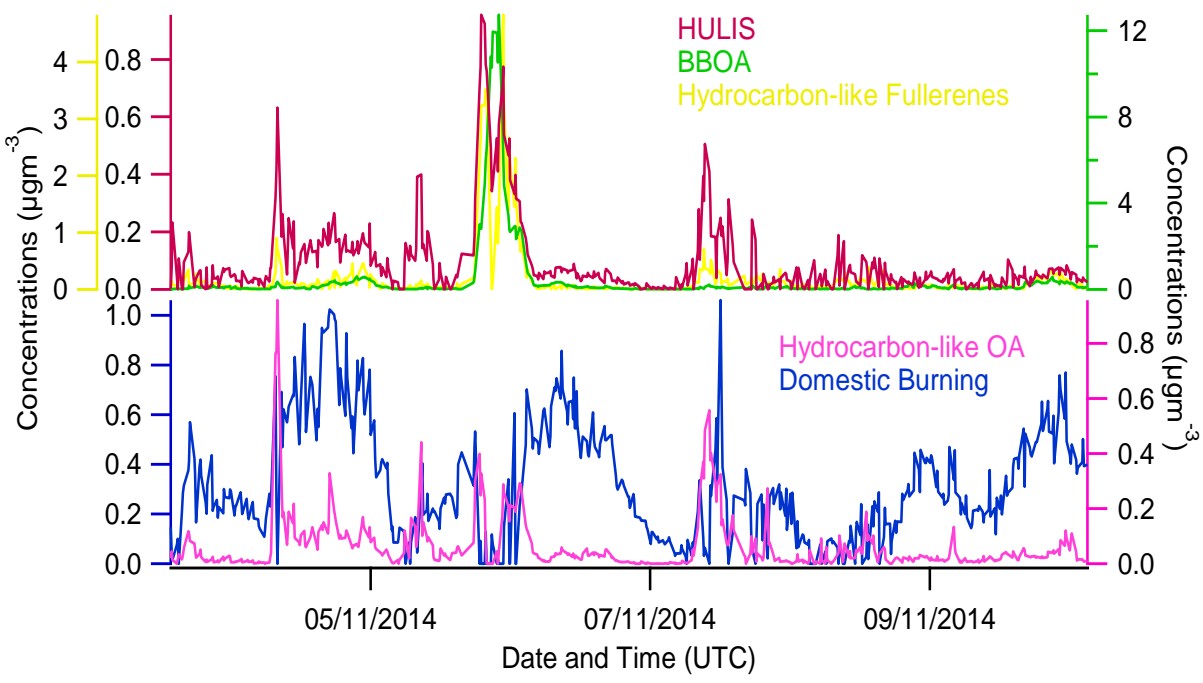

**Figure 12: The contributions of BC signals in μg/m³ after normalising it to the concentrations of BC₉₅₀ from aethalometer (AE31) (Reyes et al, 2018).**

**5 Conclusion**

This study has shown that for the first time, the inclusion of fullerene data in PMF applied to HR-SP-AMS data can be used to apportion soot into five sources. The five soot sources can be divided into bonfire night factors (HOA-like Fullerene, HULIS, BBOA) and non-bonfire night factors (i.e. domestic wood burning, hydrocarbon-like organic aerosol and some HULIS). Metals were also observed at the time of fireworks display such as Fe, Ti and Cs and Sr. But Sr was most unambiguously associated with the fireworks, due to the fact, that there is no other source of Sr signal present outside of Bonfire night. metal was taken as a tracer for the fireworks. The addition of fullerene signals and increasing the model error value from 0 to 0.10 reduces the uncertainty/error in the PMF factor solution and provides the best factorisation results. The fullerene data was successfully incorporated into rBC signals and linked with the bonfire emissions directly while Sr metal signals did not incorporate into rBC or any other factors, implying it was not contributing as a source. Also, the inclusion of Fullerene signals also helped to differentiate between different factors. The results correlate well with the other BC and soot proxies provided by other instruments presented in previous papers and can be used to estimate the relative contributions of the different





sources to total BC. This technique will be useful in the future studies to better differentiate between the different

soot sources in complex polluted environments.

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

**7 Authors Contribution**

Zainab Bibi performed the data analysis and wrote the paper. James D. Allan and Paul I. Williams designed the experiment and operated the SP-AMS. Ernesto Reyes Villegas, Michael Priestley and Carl Percival provided measurements and data from other instruments. Zainab Bibi was supervised by James D. Allan and Hugh Coe, with Ernesto Reyes Villegas and James Brooks assisting with PMF analysis.



## 8 Data Availability

Data is archived at the University of Manchester and available on request'. It will be posted publicly before submission of the final manuscript.

## 9 Acknowledgement

This work was supported by NERC grant COM-PART (NE/K014838/1) for data collection. Zainab Bibi's PhD was funded by a Dean's Award Scholarship from the University of Manchester Faculty of Science and Engineering.
