# Peer review of "Technical Note: A new approach to discriminate different Black Carbon sources by utilising Fullerene and metals in Positive Matrix Factorisation analysis of High-Resolution Soot Particle Aerosol Mass Spectrometer data"

_Atmospheric Chemistry and Physics, 2020_

## Referee Comment (RC1) · Anonymous Referee #1 · 14 Oct 2020

This paper presents a new approach by using fullerenes and metals in PMF analyses of HR-SAP-AMS data, and shows that BC from more than one biomass burning sources can be separated. The work can be a good contribution to the aerosol chemistry, however, there are quite a number of important issues to be addressed first before considering its possible publication in ACP. Please see below:

(1) This work uses fullerenes and metals to help the BC source apportionment via SP-AMS, however, it is surprised that the authors seem to completely overlook a few important papers very closely related with your current analyses. Necessary citations and incorporation of findings from these studies has to be added in your work. And I also suggest the authors to do a more thorough search on the recent SP-AMS studies. These papers at least include: Distribution of carbon cluster ions in different BC types (Aerosol Sci Tech 2015;49:409-422); detection, quantification and source apportionment of fullerenes by using SP-AMS (Environ Sci Technol Lett 2016,3,121-126); Characterization of OA via BC fragments and metals detected by SP-AMS (Atmos. Chem. Phys., 20, 5977–5993, 2020); Source identification of BC by using SP-AMS (Atmos Environ 2018,185,147-152); Another study combining SP-AMS and SP2 data to apportion BC into different sources (Atmos Chem Phys 2019,19,6749-6769)

(2) Line 115: What is the role of a catalytic stripper? And why data under this mode is not used then?

(3) Line 120:Why no ionization efficency was not obtained? Due to what difficulty? If other studies can determine the IE, then why this work cannot? It is not explained clearly to the readers. If the IE or RIE of different species, especially fullerenes, are not determined, this is a fatal issue. This reviewer doubts the credibility of PMF results and subsequent analyses.

(4) Section 2.2: Some necessary technical details are still missing here: what is the chemical resolution, and how does this allow you to detect the ions with low signals? For example, different metals? What is the time resolution? And what are the detection limits of different species by using SP-AMS? At last, did the SP-AMS here only detect BC-containing particles? Then it is not clearly specified that the SP-AMS is operated with only laser vaporizer? Or with laser and tungsten vaporizer? Is the thermal vaporizer physically removed? The different modes significantly influenced the aerosol components detected (Check and cite if necessary: Atmos. Meas. Tech. 2014, 7, 4507−4516; Atmos Chem Phys 2019,19,447-458)

(5) Section 2.3: m/z up to 3200 was determined, however, the m/z range of your PMF results is only up to 1000. Can you explain?

(6) Figure 4. More details are needed. How did you determine BrC for example?

(7) Figure 5. How to determine PON etc? A citation of previous work is not enough.

(8) Figure 6. You have a high resolution SP-AMS, why not use different colors to differentiate different ion categories? (Figure 7 is redundant). This helps justification of your PMF results and better for readers to judge and understand your results. Also, the justification of your PMF results is not clear. I am not sure for example, why factor 5 is fullerenes, and why factor 1 is BC and HULIS, and so on? The O/C and H/C are too high and they seem to be wrong. This has to be addressed well, otherwise, analyses based on the PMF are not trustworthy.

(9) How about the diurnal patterns of your factors?

(10) Figure 9 is not clear, please replace with a high resolution one

---

## Short Comment (SC1) · 27 Oct 2020

SUBJECT: Percentage uncertainty terms (peak width and fitting errors) results in a similar conclusion as PMF model error

TEXT:

Dear authors,

[Figure]

It is very interesting to see these fullerene and metal ions included in SP-AMS–PMF. I would like to make two comments.

The first comment is trivial. Figure 4 uses a confusing mixture of terminology and units. Absorption measurements are reported as both BC [$\mu$g m$^{-3}$] and BrC [/Mm]. I would suggest following the recommendations of Petzold et al. 2013 and renaming BC as eBC, for both AE31 and MAAP, and specifying in the figure legend the measurement wavelength and assumed MAC. Also, the reason that the two eBC instruments are plotted on separate axes could be noted in the legend.

The second comment is to recommend an improvement to the authors' SP-AMS error model based on Corbin et al. 2015a ("C2015") instead of using PMF's "model error" to address SNR issues. I am sure that the authors will already understand my suggestion from this brief statement. However, I would like to take the opportunity of writing this Interactive Comment to summarize C2015's message in simple practical terms:

Currently, the large dynamic range of signals in this data set led the authors to use a PMF "model error" of 0.05 or 0.10. This is illustrated by the signal to noise ratios (SNRs) in Figures S3b, S4b, S5b, which showed extremely high values for some ions. Specifically, the SNR for ions such as C3H8O was about 1000, whereas the SNR for fullerene ions was about 10 (too small to read from Figure S3b). This is a perfect example of the problems described in C2015, and arises when only Poisson counting errors are taken into account. In real-world instruments, Poisson-only error models will almost always overestimate the SNR of large signals, and this problem can be overcome with the inclusion of a percentage error term (Rocke and Lorenzato, 1995). The effect of this percentage error term is easy to understand if the SNR is converted to a percentage. SNRs of 1000 and 10 become percentages of 0.1% and 10%, respectively. Normally, repeated measurements vary by much more than 0.1%, which implies that the SNR of 1000 is an overestimate.

The PMF "model error" of 0.05 or 0.10 acts like a percentage error term (of 5% or

10% respectively). However, it has no physical basis, beyond implying that the bilinear PMF model does not represent the data accurately. In contrast, including the missing proportional error terms in the SP-AMS uncertainty model would result in nearly the same numerical outcome, but without implying a problem in the actual factorization stage or requiring a subjective decision on how much error to include. Thus, including of peak width and peak fitting errors in AMS-PMF reduces uncertainty in the final result.

Although C2015 discussed a complex Monte Carlo approach to peak-fitting errors, two of the conclusions of that work could be applied here very easily. One is that isolated peaks (no overlap) will always have a couple of percentage uncertainty due to random noise in the $m/z$ calibration (Figure 8 in C2015 implies about 2% or 3%). The other has nothing to do with peak fitting, and only regards peak integration. During peak integration (Eq. 4 in C2015), the peak width $w$ is predicted from an empirical fit to the data (often a linear fit, see Section 2.2 in C2015 and the SI of Corbin et al. 2015b). This prediction of $w$ has an associated proportional uncertainty $\sigma_w/w$. In the C2015 data set $\sigma_w/w$ was 2.5%. This 2.5% may be treated as independent of the earlier 2% or 3% uncertainty in the height of isolated peaks, so the two can be summed in quadrature. So, the total percentage uncertainty (Eq. 6 in C2015) for isolated peaks can be about 5%. This 5% is conceptually equivalent to 0.05 model error. This 5% is likely may be an underestimate for the present study, where the authors have investigated peaks at very high $m/z$ that were not addressed in C2015. (The authors might add a comment on the calibration of these peaks.)

There is only a minor technical difference between model error and total percentage uncertainty: the PMF Evaluation Tool [PET] does the sum linearly instead of quadratically.

In summary: one can expect a few percent of isolated-peak uncertainty in all cases (due to variability in m/z calibration) and another few percent of peak-integration uncertainty ($\sigma_w/w$). The latter can be estimated from the peak-width calibration stage in PIKA (by repeating the peak width fit manually, as detailed in Corbin et al. 2015b). This

provides a simple and objective justification for using model error in PMF.

Again, although I have taken the opportunity to write this out in full, I am sure that the authors understand these details, and that they only add weight to the results of Bibi et al. 2020.

Best regards,

Joel

**References**

Corbin, J. C, Othman, A., D. Haskins, J., D. Allan, J., Sierau, B., R. Worsnop, D., Lohmann, U. and A. Mensah, A.: Peak fitting and integration uncertainties for the Aerodyne Aerosol Mass Spectrometer, Atmos. Meas. Tech. Discuss., 8(4), 3471–3523, doi:10.5194/amtd-8-3471-2015, 2015a. *"C2015"*

Corbin, J. C., Lohmann, U., Sierau, B., Keller, A., Burtscher, H. and Mensah, A. A.: Black carbon surface oxidation and organic composition of beech-wood soot aerosols, Atmos. Chem. Phys., 15(20), 11885–11907, doi:10.5194/acp-15-11885-2015, 2015b.

Petzold, A., Ogren, J. A., Fiebig, M., Laj, P., Li, S.-M., Baltensperger, U., Holzer-Popp, T., Kinne, S., Pappalardo, G., Sugimoto, N., Wehrli, C., Wiedensohler, A. and Zhang, X.-Y.: Recommendations for the interpretation of "black carbon" measurements, Atmos. Chem. Phys., 13(16), 8365–8379, doi:10.5194/acp-13-8365-2013, 2013.

Rocke, D. M. and Lorenzato, S.: A two-component model for measurement error in analytical chemistry, Technometrics, 37(2), 176–184, doi:10.1080/00401706.1995.10484302, 1995.

---

## Referee Comment (RC2) · Anonymous Referee #2 · 6 Dec 2020

General comments:

This work attempts to integrate fullerene signals detected by the high-resolution soot-particle aerosol mass spectrometer (HR-SP-AMS) for source apportionment of ambient black carbon (BC) in an urban environment during the period with the bonfire event. The proposed positive matrix factorization (PMF) analysis has great potential to advance our understanding on the sources of OA and BC emitted from different

combustion processes that cannot be easily resolved from conventional HR-AMS measurements.

As a technical note of ACP, my major concern is that this work did not demonstrate clearly how the additional fullerene signals can improve (1) the fundamental understanding (e.g., sources, transport, chemistry, etc.) in their case study and/or (2) black carbon source apportionment as the total fullerene signals are weak in general. These possibly can be achieved by comparing different PMF scenarios (e.g., including BC fragments in PMF with vs. without fullerene signals to see if different mass spectral profiles or number of factors may be obtained) if the results are available. I also have a few suggestion to improve the presentation quality of this work. Overall, the above general comments should be addressed together with the specific comments below in the revised version before considering to be published in ACP.

Major comments:

1. Introduction: Although the proposed method includes a new concept for data analysis, this manuscript should better recognize the contribution of other recent SP-AMS studies that performed fullerene detection near sources and that integrated BC signals in PMF for BC and OA source analysis.

2. Instrumentation: HR-SP-AMS has been deployed in many field studies with different configurations and operation modes. I do believe the tungsten vaporizer was removed from the instrument in this work as only BC-containing particles were detected. It would be very beneficial to readers who are not familiar with this instrument if the authors can explicitly describe whether the HR-SP-AMS was operated in the presence or absence of tungsten vaporizer and what can be detected with this specific configuration.

3. Lines 126-127: As there were only limited work to report metal detection in ambient particles using HR-SP-AMS, It is recommended to include a list of metal peaks that have been investigated and/or detected in this work.

4. Lines 131-134: Please define fullerene peaks. If UMR data is used for large m/z, what is the possible error for determining fullerene signals?

5. Lines 137-138. There are a few previous field studies that included BC fragments for source apportionment/identification analysis of ambient BC and OA but they may not explicitly highlight this application in the manuscript. However, those publications should be cited here.

6. Figures: Although Figure 4 is good for visualization, it is recommended to report the Pearson correlation coefficients between different BC measurements here. This comment also applies to other time series comparisons throughout the manuscript. Furthermore, Figures 2-5 can be combined into a single graph with different panels so that the time series of different species can be easily compared. Figures 6 and 7 can be combined as well (i.e., showing HR-MS for lower range m/z in Figure 6).

7. Section 3.5: Re-organization of this section is required. In particular, it is recommended to discuss the PMF factor profile and time series together instead of separating them into two sub-section as both of them provide information for sources of BC and OA. For example, Figure 8 (time series of PMF factors) is required at the beginning of Section 3.5 when describing which OA factors were strongly associated with the bonfire night or other emissions. The mass spectral profile alone did not provide sufficient evidence to support the scientific argument.

8. Line234: The meaning of HULIS here is unclear. Figure 6 only shows "BC and HULIS" factor. The terminology throughout the manuscript should be consistent.

9. Lines 243-245: Both factors 2 and 3 consist of fullerene peaks. Please further elaborate how the fullerenes help to differentiate domestic burning and biomass burning during the bonfire event (e.g., any distinct peaks or mass spectral pattern that can be used?). It seems that the lower m/z fragments are more than sufficient to tell the differences between the two OA factors. What does "hydrocarbon like fullerene" mean?

10. Lines 287-288: Three bonfire night factors were identified. Were they all from bonfire emissions? If so, it implies that there were different types of bonfire emissions that can provide sufficient temporal variabilities for PMF factor separation. I am wondering if the same number of PMF factors can be obtained if fullerene signals is excluded. More discussion is required to demonstrate the importance of including fullerene signals in PMF analysis.

11. HULS factor: The manuscript mention a couple of times that a factor having strong m/z 44 signals can represent HULIS in ambient particles, but I cannot fully follow the flow of argument. My interpretation is that the mass spectral features of the HULIS factor is similar to that of more-oxidized oxygenated OA (MO-OOA) factor identified in most other field studies. I am wondering whether other co-located measurements in this work can provide evidence that the HULIS factor has some specific chemical features that cannot be described as MO-OOA. I understand this can be just a terminology issue. More elaboration is required here.

Minor comments: 1. Line 280. I think m/z 73 instead of m/z 71 for typical biomass burning factors. 2. Line 337: Please define BCtr. 3. Line 345: Please define BCwb.

---

## Author Comment (AC1) · 28 Feb 2021

This paper presents a new approach by using fullerenes and metals in PMF analyses of HR-SP-AMS data and shows that BC from more than one biomass burning sources can be separated. The work can be a good contribution to the aerosol chemistry, however, there are quite a number of important issues to be addressed first before considering its possible publication in ACP. Please see below:

[Figure]

(1) This work uses fullerenes and metals to help the BC source apportionment via SPAMS, however, it is surprised that the authors seem to completely overlook a few important papers very closely related with your current analyses. Necessary citations and incorporation of findings from these studies has to be added in your work. And I also suggest the authors to do a more thorough search on the recent SPAMS studies. These papers at least include: Distribution of carbon cluster ions in different BC types (Aerosol Sci Tech 2015;49:409-422); detection, quantification and source apportionment of fullerenes by using SP-AMS (Environ Sci Technol Lett 2016,3,121-126); Characterization of OA via BC fragments and metals detected by SP-AMS (Atmos. Chem. Phys., 20, 5977–5993, 2020); Source identification of BC by using SP-AMS (Atmos Environ 2018,185,147-152); Another study combining SP-AMS and SP2 data to apportion BC into different sources (Atmos Chem Phys 2019,19,6749-6769).

Ans: Thank you so much for your feedback and suggesting me some new research articles to include in my journal. I will consider your suggestion and add these citation work in my manuscript.

Onasch, T.B., Fortner, E.C., Trimborn, A.M., Lambe, A.T., Tiwari, A.J., Marr, L.C., Corbin, J.C., Mensah, A.A., Williams, L.R., Davidovits, P. and Worsnop, D.R., 2015. Investigations of SP-AMS carbon ion distributions as a function of refractory black carbon particle type. Aerosol Science and Technology, 49(6), pp.409-422.

Wang, J., Onasch, T.B., Ge, X., Collier, S., Zhang, Q., Sun, Y., Yu, H., Chen, M., PreÌĄvoÌĆt, A.S. and Worsnop, D.R., 2016. Observation of fullerene soot in eastern China. Environmental Science & Technology Letters, 3(4), pp.121-126.

Carbone, S., Onasch, T., Saarikoski, S., Timonen, H., Saarnio, K., Sueper, D., Rönkkö, T., Pirjola, L., Worsnop, D. and Hillamo, R., 2015. Characterization of trace metals with the SP-AMS: detection and quantification. Atmospheric Measurement Techniques Discussions, 8(6).

Liu, D., Joshi, R., Wang, J., Yu, C., Allan, J.D., Coe, H., Flynn, M.J., Xie, C., Lee, J.,

Squires, F. and Kotthaus, S., 2019. Contrasting physical properties of black carbon in urban Beijing between winter and summer. Atmospheric Chemistry and Physics, pp.6749-6769.

(2) Line 115: What is the role of a catalytic stripper? And why data under this mode is not used then?

Ans: Thanks for asking. The catalytic stripper was used in the setup and has already published in Danton Liu's paper. It was not included for this research work publication. The only reason to quote in my manuscript is, it's been used in another study. I have also tried analysing catalytic stripper data but did not get sensible results. So, I have excluded that data.

(3) Line 120: Why no ionization efficiency was not obtained? Due to what difficulty? If other studies can determine the IE, then why this work cannot? It is not explained clearly to the readers. If the IE or RIE of different species, especially fullerenes, are not determined, this is a fatal issue. This reviewer doubts the credibility of PMF results and subsequent analyses.

Ans: Thank you so much for your comment. At the time, we tried but it did not work properly so we cannot go back and recalibrate. Also, RIE did not affect PMF analysis results and it's not at all fatal issues.

(4) Section 2.2: Some necessary technical details are still missing here: what is the chemical resolution, and how does this allow you to detect the ions with low signals? For example, different metals? What is the time resolution? And what are the detection limits of different species by using SP-AMS? At last, did the SP-AMS here only detect BC-containing particles? Then it is not clearly specified that the SP-AMS is operated with only laser vaporizer? Or with laser and tungsten vaporizer? Is the thermal vaporizer physically removed? The different modes significantly influenced the aerosol components detected (Check and cite if necessary: Atmos. Meas. Tech. 2014, 7, $4507-4516$; Atmos Chem Phys 2019,19,447-458)

Ans: In terms of technical details, High Resolution SPAMS detected the metal signals such as Sr, Fe, Cs and Ti. The time resolution, as I have mentioned in my manuscript was from 29th Oct-11th Nov 2014 (line 94). Also, I have clearly stated that in this study, the SP-AMS was only operated with the laser vaporiser only in combination with the electron impact source and measures both refractory and nonrefractory components along with metal nanoparticles and it's also cited (Onasch et al., 2012).

(5) Section 2.3: m/z up to 3200 was determined, however, the m/z range of your PMF results is only up to 1000. Can you explain?

Ans: Thank you so much for your comment. The reason for 1000 m/z range of this PMF analysis is that above m/z 1000 the signals were not approachable and it was only noise so that's why we have ignored them and only added m/z up to 1000 by clearly showing the separate BC sources.

(6) Figure 4. More details are needed. How did you determine BrC for example?

Ans: The Aethalometer AE31 was used to determine BrC and it has already published in Reyes et al., 2018.

(7) Figure 5. How to determine PON etc? A citation of previous work is not enough.

Ans: Particulate organic oxides of nitrogen (PONs) were estimated using the m/z 46 : 30 ratios from aerosol mass spectrometer (AMS) measurements, according to previously published methods (published in Reyes et al., 2018).

(8) Figure 6. You have a high-resolution SP-AMS, why not use different colors to differentiate different ion categories? (Figure 7 is redundant). This helps justification of your PMF results and better for readers to judge and understand your results. Also, the justification of your PMF results is not clear. I am not sure for example, why factor 5 is fullerenes, and why factor 1 is BC and HULIS, and so onïïj§ The O/C and H/C are too high, and they seem to be wrong. This has to be addressed well, otherwise, analyses based on the PMF are not trustworthy.

Ans: Thank you so much for the suggestion. Sure, I will use different colours to differentiate the ion categories. I have added some more details to explain why factor 1 is BC and HULIS and others as well. Yes, you are right, I have mistakenly added the wrong values of O:C but H:C values are correct. Thank you so much for pointing this, I have now added the correct values.

(9) How about the diurnal patterns of your factors?

Ans: The data for this study is only for few days, and it is already obvious from the time series that the signals were very high during the bonfire night specifically, when the fire log burning, and fireworks started. And there is no point of doing diurnal plots because it is obvious that it will be dominated by bonfire night.

(10) Figure 9 is not clear, please replace with a high resolution one Ans: Thank you so much for pointing this out. I have replaced figure 9 in manuscript.

Please also note the supplement to this comment:
https://acp.copernicus.org/preprints/acp-2020-890/acp-2020-890-AC1-supplement.pdf

———————————————————

[Figure]

**Fig. 1.** PMF five factors source profile (factor 1 = BC and HULIS, factor 2 = BBOA, factor 3 = Domestic burning OA, factor 4 = Hydrocarbon-Like OA, factor 5 = Fullerene). Note the difference in scales of the f

[Figure]

**Fig. 2.** The similarity between different pollutant time series through hierarchical cluster analysis (HCA)

**Supplement:**

**Supplementary Materials**

**Table S1: Pearson correlation coefficients between different BC measurements such as BC (HR-SP-AMS) with BC and BrC (AE31) and BC (MAAP)**

| BC (HR-SP-AMS) | |
|---|---|
| | Pearson Coefficient |
| BC (AE31) | 0.98 |
| BrC (AE31) | 0.96 |
| BC (MAAP) | 0.95 |

**Table S2: Correlation between BC (HR-SP-AMS) and CIMS measurements**

| HR-SP-AMS | CIMS DATA | | |
|---|---|---|---|
| | HCN | HCNO | HONO |
| | Pearson Coefficient | Pearson Coefficient | Pearson Coefficient |
| rBC (HR-SP-AMS) | 0.88 | 0.77 | 0.89 |

**Table S3: Correlation between HR-Aerosols species Vs Aerosol and Gases (AMS)**

| HR Aerosol Species | Aerosol and Gases | Pearson Coefficient |
|---|---|---|
| rBC | BC_(ugm$^{-3}$) | 0.95 |
| HROrg | Org_(ugm$^{-3}$) | 0.92 |
| HRNH$_4$ | NH$_4$_(ugm$^{-3}$) | 0.92 |
| HRNO$_3$ | NO$_3$_(ugm$^{-3}$) | 0.86 |
| HRSO$_4$ | SO$_4$_(ugm$^{-3}$) | 0.91 |
| HRChl | Chl_(ugm$^{-3}$) | 0.99 |

**PMF Factorisation factors solution without inclusion of Fullerenes signals:**

[Figure]

**Figure S1a Mass Spectra of five factor solution (without inclusion of fullerene signals).**

[Figure]

**Figure S1b time series of five factors (without the inclusion of fullerene data).**

**PMF Factorisation factors solution with inclusion of Fullerenes signal**

[Figure]

**Figure S2a: PMF five factors source profile (factor 1 = BC and HULIS, factor 2 = BBOA, factor 3 = Domestic burning OA, factor 4 = Hydrocarbon-Like OA, factor 5 = Fullerene). Note the difference in scales of the fullerene signals (right hand axes).**

[Figure]

**Figure S2b: The time series of (a) non-Bonfire night factors and (b) bonfire night factors obtained by PMF analysis of the HR-SP-AMS data.**

**Model Error modification:**

Here we use the term 'model' error to refer to the additional error term that can be added as part of the PET toolkit, whereby additional error is added proportional to the signal, as opposed to the square root of the signal as is done

in the standard AMS error model. In this study, the model error parameter was modified from 0.00, 0.05 and 0.10 by following the recommendations made by Paatero and Hopke (2003). While this is not always done in AMS PMF analysis, this is done here to decrease the signal-to-Noise Ratio (SNR) of the high signals that would otherwise dominate the factorisation. Firstly, the PMF was run with 0.0 model error and Fig. S3a, b depicts that SNR was very high i.e. 1200 SNR for some peaks. For a model error parameter of 0.05, fig. S4a, b shows that SNR was decreased i.e. 18 SNR and improved overall signals of sources, particularly for the fullerenes. For a value of 0.10, the maximum SNR is further reduced to 10 (fig. S5a, b), meaning the error was now dominated by the 'model error' term, and this delivered the most satisfactory solution in terms of distinctive mass spectra, particularly for the fullerenes.

[Figure]

**Fig S3a PMF five factors profile detected by the model error 0.0**

[Figure]

**Figure S3b shows the SNR of organics and fullerenes with no modification in the model error value i.e. 0.00**

[Figure]

**Fig S4a PMF five factors profile detected by the model error 0.05**

[Figure]

**Figure S4b shows the SNR of organics and fullerenes with little modification in the model error value i.e. 0.05**

[Figure]

**Fig S5a PMF five factors profile detected by the model error 0.10**

[Figure]

**Figure S5b shows the SNR of organics and fullerenes with more modification in the model error value i.e. 0.10.**

**PMF factors-solution selection:**

A range of solutions were obtained using different parameters as part of the PMF analysis and here we present the reasons behind the choice of solution used in the paper. Regarding the number of factors, a 5-factor solution was chosen instead of 6-factor solution because all the five factors are separated from one another and represent a specific soot source (fig S6a, b). In comparison, the 6-factor solution has two 'split' factors representing the same

emissions. These are factor 2 and factor 4 in figure S7b and represent domestic wood burning sources because their peaks were evident before and after the bonfire night event (fig. S7a, S7b).

[Figure]

**Figure S6a: Time series of five factors solution detected separately under the condition of 0.10 model error.**

[Figure]

**Figure**

**S6b: The factor profile of five factors solution detected separately under the condition of 0.10 model error.**

[Figure]

**Figure S7a: Time series of six factors solution in which two same factors are split in to two different factors i.e. 2 and 4 under the condition of 0.10 model error.**

[Figure]

**Figure S7b: Factor profiles of the 6-factor solution in which factor 2 and 4 have the same m/z spectrum, under the condition of 0.10 model error.**

---

## Author Comment (AC2) · 28 Feb 2021

1. Introduction: Although the proposed method includes a new concept for data analysis, this manuscript should better recognize the contribution of other recent SP-AMS studies that performed fullerene detection near sources and that integrated BC signals in PMF for BC and OA source analysis.

I have added some recent SP-AMS studies from the following references which stated

the detection of fullerenes near sources and integrated BC signals by applying PMF for OA and BC sources.

Wang, J., Onasch, T.B., Ge, X., Collier,wang S., Zhang, Q., Sun, Y., Yu, H., Chen, M., Prevot, A.S. and Worsnop, D.R., 2016. Observation of fullerene soot in eastern China. Environmental Science & Technology Letters, 3(4), pp.121-126.

Carbone, S., Onasch, T., Saarikoski, S., Timonen, H., Saarnio, K., Sueper, D., Rönkkö, T., Pirjola, L., Worsnop, D. and Hillamo, R., 2015. Characterization of trace metals with the SP-AMS: detection and quantification. Atmospheric Measurement Techniques Discussions, 8(6).

Liu, D., Joshi, R., Wang, J., Yu, C., Allan, J.D., Coe, H., Flynn, M.J., Xie, C., Lee, J., Squires, F. and Kotthaus, S., 2019. Contrasting physical properties of black carbon in urban Beijing between winter and summer. Atmospheric Chemistry and Physics, pp.6749-6769.

Onasch, T.B., Fortner, E.C., Trimborn, A.M., Lambe, A.T., Tiwari, A.J., Marr, L.C., Corbin, J.C., Mensah01, A.A., Williams, L.R., Davidovits, P. and Worsnop, D.R., 2015. Investigations of SP-AMS carbon ion distributions as a function of refractory black carbon particle type. Aerosol Science and Technology, 49(6), pp.409-422.

2. Instrumentation: HR-SP-AMS has been deployed in many field studies with different configurations and operation modes. I do believe the tungsten vaporizer was removed from the instrument in this work as only BC-containing particles were detected. It would be very beneficial to readers who are not familiar with this instrument if the authors can explicitly describe whether the HR-SP-AMS was operated in the presence or absence of tungsten vaporizer and what can be detected with this specific configuration.

I have added these few lines in introduction chapter. Thank you for pointing this. In this study, the HR-SP-AMS used was not the same as the C-ToF-AMS (Compact Time-of-Flight Aerosol Mass Spectrometer) described in Reyes et al., (2018). The HR-SP-AMS

was operated under an intracavity, CW laser vaporiser (with the tungsten vaporiser removed), which vaporises the refractory BC (rBC) and its associated non-refractory particulate species along with metal nanoparticles (Onasch et al., 2012; Carbone et al., 2015).

3. Lines 126-127: As there were only limited work to report metal detection in ambient particles using HR-SP-AMS, it is recommended to include a list of metal peaks that have been investigated and/or detected in this work.

Thank for your comment. In line 183, I have mentioned four different metals that have been detected by HR-SP-AMS, please see below: "This high-resolution analysis on SP-AMS data also detected various metal pollutants such as Iron (Fe), Titanium (Ti), Strontium (Sr) and Caesium (Cs)".

4. Lines 131-134: Please define fullerene peaks. If UMR data is used for large m/z, what is the possible error for determining fullerene signals?

Thank you so much for your suggestion regarding this comment, I have improved it now; This lower frequency delivered data up to m/z=3200 rather than 380, with the intention of characterising the fullerene signals described by Onasch et al. (2012) at the expense of overall signal-to-noise. The data presented in this paper are a combination of the standard 'V' mode for the lower m/z peaks, processed using the PIKA high resolution analysis tool, and the long pulser period 'V' mode for the fullerene peaks, processed using unit mass resolution (UMR) method. The reason for using UMR method instead of HR was that the peaks in this m/z regime were not sufficiently resolved, due to the m/$\Delta$m limit of the mass spectrometer. Instead, the UMR method can integrates all the available signals and is therefore more robust. However, the ability to resolve multiple peaks per nominal integer m/z provided useful additional data in the low m/z regime. The 'W' mode data was deemed not to have a sufficient signal-to-noise ratio to contribute to this work.

5. Lines 137-138. There are a few previous field studies that included BC fragments for source apportionment/identification analysis of ambient BC and OA but they may not explicitly highlight this application in the manuscript. However, those publications should be cited here. Thanks for your suggestion. This citation is added in the manuscript. Saarikoski, S., Carbone, S., Cubison, M.J., Hillamo, R., Keronen, P., Sioutas, C., Worsnop, D.R. and Jimenez, J.L., 2014. Evaluation of the performance of a particle concentrator for online instrumentation. Atmospheric Measurement Techniques, 7(7), pp.2121-2135.

6. (a) Figures: Although Figure 4 is good for visualization, it is recommended to report the Pearson correlation coefficients between different BC measurements here. This comment also applies to other time series comparisons throughout the manuscript.

Thank you so much for this suggestion. I will add following tables in the supplementary information section.

6b. Furthermore, Figures 2-5 can be combined into a single graph with different panels so that the time series of different species can be easily compared. Figures 6 and 7 can be combined as well (i.e., showing HR-MS for lower range m/z in Figure 6).

Thanks for your suggestion, I have combined figures 2, 3, 4 and 5 in one panel.

7. Section 3.5: Re-organization of this section is required. In particular, it is recommended to discuss the PMF factor profile and time series together instead of separating them into two sub-section as both of them provid3e information for sources of BC and OA. For example, Figure 8 (time series of PMF factors) is required at the beginning of Section 3.5 when describing which OA factors were strongly associated with the bonfire night or other emissions. The mass spectral profile alone did not provide sufficient evidence to support the scientific argument.

Thank you so much for your suggestion. I have reorganised this section now (See revised manuscript).

8. Line 234: The meaning of HULIS here is unclear. Figure 6 only shows "BC and

HULIS" factor. The terminology throughout the manuscript should be consistent.

In order to provide the clearer meaning of HULIS, I am adding further explanation to this; HULIS is a class of organic molecules that can be formed by photochemical oxidation and oligomerisation of volatile organic compounds in the atmosphere (Aiken et al., 1985; Hoffer et al., 2004) and biomass burning (Lin et al., 2010), with a characteristic peak at m/z 44 (McFiggans et al, 2005). Potential origins of HULIS in the atmosphere are diverse, including (primary) biomass burning (Graber and Rudich, 2006; McFiggans et al, 2005; Mukai and Ambe, 1986; Zappoli et al., 1999; Graham et al., 2002; Mayol-Bracero et al., 2002), terrestrial (Simoneit, 1980) and marine sources (Cini et al., 1994; Cini et al., 1996; Calace et al., 2001; Cavalli et al., 2004), , and secondary organic aerosol formation (condensation, reaction, oligomerisation, etc.) (Gelencser et al., 2002; Jang et al., 2002; Jang et al., 2003; Tolocka et al., 2004; Hung et al., 2005). Moreover, HULIS as an atmospheric aerosol has already been reported in previous literature (Decesari et al., 2000, 2007). Along with this the work of Havers et al. (1998), wherein the term HULIS was coined. Examining a standard reference air dust as well as airborne particulate matter, Havers et al. (1998) attributed 10% or more of aerosol organic carbon to macromolecular substances HULIS similar to humic and fulvic acids. Aiken, G.R., McKnight, D.M. and Wershaw, R.L. (1985). Humic substances in soil, sediment, and water. Geochemistry, Isolation and Characterization. New York: Wiley

9. Lines 243-245: Both factors 2 and 3 consist of fullerene peaks. Please further elaborate how the fullerenes help to differentiate domestic burning and biomass burning during the bonfire event (e.g., any distinct peaks or mass spectral pattern that can be used?). It seems that the lower m/z fragments are more than sufficient to tell the differences between the two OA factors. What does "hydrocarbon like fullerene" mean?

Both factors 2 and 3 have fullerene peaks but if we see their y-axis the concentration of fullerene is very low and mainly dominated by other sources, while factor 1 is heavily populated by fullerene peaks, so that's why I have only considered factor 1 as fullerene (Onasch et al., 2015). Although it is not clear why fullerene signals are sometimes observed, it does seem to differentiate between biomass burning during the bonfire event and biomass burning from domestic burning. In fig 4, Factor 5 was heavily weighted by hydrocarbon like Fullerene having a peak at m/z 720 (C60+), implying polycyclic aromatic hydrocarbons can transform into soot containing Fullerene during combustion (Wang et al., 2015, Wang et al., 2016; Reilly et al., 2000). This was typically not associated from the traffic source (diesel), so depended on the different type of combustion. References: Reilly, P. T. A.; Gieray, R. A.; Whitten, W. B.; Ramsey, J. M. Fullerene Evolution in Flame-Generated Soot. J. Am. Chem. Soc. 2000, 122 (47), 11596−11601. Wang, J., Onasch, T.B., Ge, X., Collier, S., Zhang, Q., Sun, Y., Yu, H., Chen, M., PreÌĄvoÌĆt, A.S. and Worsnop, D.R., 2016. Observation of fullerene soot in eastern China. Environmental Science & Technology Letters, 3(4), pp.121-126.

10. Lines 287-288: Three bonfire night factors were identified. Were they all from bonfire emissions? If so, it implies that there were different types of bonfire emissions that can provide sufficient temporal variabilities for PMF factor separation. I am wondering if the same number of PMF factors can be obtained if fullerene signals is excluded. More discussion is required to demonstrate the importance of including fullerene signals in PMF analysis.

All of the three factors are bonfire emissions factors (please see the time series spike originated during the bonfire emissions time period).

Factorisation without Fullerenes: Thank you so much for pointing out this comment. I have added the factorisation without fullerene graphs in the supplementary section and their explanation in the result section. Please see below the factorisation results. Firstly, the factorisation was performed without the inclusion of fullerene signals in the data matrix, in order to explore the factorisation without fullerene data. And the results showed five factors solution (fig S1a and S1b) which are BC and HULIS, SV-OOA, BBOA, Hydrocarbon-like OA and domestic burning. In that case, only two unambiguously bonfire night sources of BC were identified, with a degree of 'mixing' between the bonfire night factor and traffic noted in the HOA factor. Also, the SV-OOA and domestic

burning factors also exhibit mixing in their timeseries as well. As such, the factorisation without fullerene signals was judged to be poor.

11. HULIS factor: The manuscript mention a couple of times that a factor having strong m/z 44 signals can represent HULIS in ambient particles, but I cannot fully follow the flow of argument. My interpretation is that the mass spectral features of the HULIS factor is similar to that of more-oxidized oxygenated OA (MO-OOA) factor identified in most other field studies. I am wondering whether other co-located measurements in this work can provide evidence that the HULIS factor has some specific chemical features that cannot be described as MO-OOA. I understand this can be just a terminology issue. More elaboration is required here.

The first factor is BC and HULIS. Firstly, by observing the time series of this factor, the reader will clearly notice that the highest concentration is during the bonfire emissions. If I interpret m/z 44 as MO-OOA then time series results also show some spikes before and after the bonfire event. Secondly, HULIS is a class of organic molecules that can be formed by photochemical oxidation and oligomerisation of volatile organic compounds in the atmosphere (Aiken et al., 1985; Hoffer et al., 2004) and biomass burning (Lin et al., 2010), with a characteristic peak at m/z 44 (McFiggans et al, 2005). Potential origins of HULIS in the atmosphere are diverse, including (primary) biomass burning (Graber and Rudich, 2006; McFiggans et al, 2005; Mukai and Ambe, 1986; Zappoli et al., 1999; Graham et al., 2002; Mayol-Bracero et al., 2002), terrestrial (Simoneit, 1980) and marine sources (Cini et al., 1994; Cini et al., 1996; Calace et al., 2001; Cavalli et al., 2004), , and secondary organic aerosol formation (condensation, reaction, oligomerisation, etc.) (Gelencser et al., 2002; Jang et al., 2002; Jang et al., 2003; Tolocka et al., 2004; Hung et al., 2005). Moreover, HULIS as an atmospheric aerosol has already been reported in previous literature (Decesari et al., 2000, 2007). Along with this the work of Havers et al. (1998), wherein the term HULIS was coined. Examining a standard reference air dust as well as airborne particulate matter, Havers et al. (1998) attributed 10% or more of aerosol organic carbon to macromolecular

substances HULIS similar to humic and fulvic acids.

Minor comments: 1. Line 280. I think m/z 73 instead of m/z 71 for typical biomass burning factors. 2. Line 337: Please define BCtr. 3. Line 345: Please define BCwb. Thank you so much for pointing out these minor mistakes. m/z 73 instead of m/z 71 is corrected eBCwb is the equivalent Black carbon emitted from wood-burning sources. eBCtr is the equivalent Black Carbon emitting from traffic emissions.

Please also note the supplement to this comment:
https://acp.copernicus.org/preprints/acp-2020-890/acp-2020-890-AC2-supplement.pdf
* * *
[Figure]

**Table S1: Pearson correlation coefficients between different BC measurements such as BC (HR-SP-AMS) with BC and BrC (AE31) and BC (MAAP)**

| BC (HR-SP-AMS) | |
| --- | --- |
| | Pearson Coefficient |
| BC (AE31) | 0.98 |
| BrC (AE31) | 0.96 |
| BC (MAAP) | 0.95 |

**Table S2: Correlation between BC (HR-SP-AMS) and CIMS measurements**

| HR-SP-AMS | CIMS DATA | | |
| --- | --- | --- | --- |
| | HCN | HCNO | HONO |
| | Pearson Coefficient | Pearson Coefficient | Pearson Coefficient |
| rBC (HR-SP-AMS) | 0.88 | 0.77 | 0.89 |

**Table S3: Correlation between HR-Aerosols species Vs Aerosol and Gases (AMS)**

| HR Aerosol Species | Aerosol and Gases | Pearson Coefficient |
| --- | --- | --- |
| rBC | BC_(ugm$^{-3}$) | 0.95 |
| HROrg | Org_(ugm$^{-3}$) | 0.92 |
| HRNH$_4$ | NH$_4$_(ugm$^{-3}$) | 0.92 |
| HRNO$_3$ | NO$_3$_(ugm$^{-3}$) | 0.86 |
| HRSO$_4$ | SO$_4$_(ugm$^{-3}$) | 0.91 |
| HRChl | Chl_(ugm$^{-3}$) | 0.99 |

**Fig. 1.** Table 1 shows the Pearson correlation coefficients between different BC measurements

[Figure]

**Figure 2. Timeseries of different variables observed during bonfire event. 2a. Time series of various metal pollutant concentrations, 2b. Time series of High Resolution rBC concentrations and its coating species (Organics and Inorganics), 2c. Time series of Black Carbon measured by different instruments i.e HR SP-AMS (rBC), AE31 (eBC and BrC) and MAAP (eBC), 2d. Time series of rBC, primary (pPON) and secondary (sPON) organic nitrate.**

**Fig. 2.** Timeseries of different variables observed during bonfire event. 2a. Time series of various metal pollutant concentrations, 2b. Time series of High Resolution rBC concentrations and its coating specie

[Figure]

[Figure]

**Fig. 3.** Mass Spectra of five factor solution (without inclusion of fullerene signals).

[Figure]

**Fig. 4.** Time series of five factors (without the inclusion of fullerene data).

---

## Author Response (AR2)

**Major Comments**

1) Both referees suggested that further citations of previous work are required. Thank you for the incorporating several of the suggested references. However, I agree with referee 2 that this discussion should be included in the introduction, rather than scattered through the paper. Please add this discussion of previous SP-AMS measurements that integrate BC signals into the PMF in the introduction and highlight the differences of your approach. An additional study that you should consider including is Rivellini et al., (2020).

I have added the further citations of previous studies detail in the introduction section.

2) Re: Referee 2 comment 11: Same as the referee, I am unable to follow the argument that supports the use of the term HULIS to describe this factor and I also wonder if MO-OOA would be a better description. While I understand that HULIS has been identified in atmospheric aerosol, in my understanding it is essentially an operational definition (similar to various measurements of BC and MO-OOA) that varies depending on the analytical instrumentation used. Moreover, HULIS generally refers to higher molecular weight species. As AMS measurements provide extremely limited information on molecular weight, I tend to agree with the referee that a terminology such as MO-OOA that has been defined based on AMS measurements is more appropriate. As stated by the referee, identification of this factor as HULIS requires "…other co-located measurements in this work can provide evidence that the HULIS factor has some specific chemical features that cannot be described as MO-OOA."

Thank you so much for your suggestion. I have replaced HULIS with MO-OOA factor. Please see the main manuscript.

3) It strikes me that the mass spectra of the fullerene region differ significantly between the different factors. Does this provide any additional insight/information particularly when compared with the Onasch et al., (2015) results?

Yes, the mass spectra of HOA+Fullerene region have different signal concentrations between different factors. But the fifth factor is named 'HOA+fullerene' because it is heavily weighted by HOA and Fullerene showing a peak at m/z 720 ($C_{60}^{+}$), implying polycyclic aromatic hydrocarbons can transform into soot containing HOA+Fullerene during combustion (Wang et al., 2015, Wang et al., 2016; Reilly et al., 2000). The mass spectra of HOA+Fullerene region have different signal concentrations between different factors (3a). But this factor is considered as HOA+Fullerene due to its high signal strength and most prominent distributions i.e., $60\times10^{-6}$ as compared to other factors (hydrocarbon-Like OA = $2.5\times10^{-6}$, domestic burning = $2.5\times10^{-6}$, BBOA = $8\times10^{-6}$ and BC and MO-OOA = $3.0\times10^{-6}$). Depending on the different phase of combustion, HOA+Fullerene was typically not associated from the traffic source (diesel), Although the mass spectral pattern varies, the signals of most factors are too low to get any meaningful information with any confidence. However, in Onasch et al., 2015, the variations within mass spectral region of fullerene signals between different factors is negligible, but in our case all the factors except 'HOA+Fullerene' factor have very low signals concentration, so this factorisation technique provide obvious and separate HOA+Fullerene factor. The time series of HOA+Fullerene also show the very high concentration of 8000 Counts/s only during the bonfire event, that is why it is the bonfire source (3b).

4) Re: Referee 1 comment 3: I agree with the referee that further information regarding the impacts of the uncertain RIE is required. How was it determined that the RIE did not affect the PMF analysis? Please describe this in the manuscript.

The RIE, as defined by Allan et al. (2004, doi: 10.1016/j.jaerosci.2004.02.007), is a constant factor applied to the signals as part of the conversion from a signal in the mass spectrometer to an ambient mass concentration. Because this is a purely multiplicative operation, this will affect all data and associated errors equally and therefore the factors derived in the PMF model described by Ulbrich et al. (2009) will simply by multiplied by the exact same amount. So, when PMF factors are derived using data that has not had an RIE applied (and corollary to this, other multiplicative factors such as ionisation efficiency, collection efficiency and inlet flow rate), the relative contributions of the different factors as a function of time will be exactly the same as if data that had had this applied. The only difference is that the absolute units of the factors are as an arbitrary mass spectral response (in s-1) rather than an ambient mass concentration (in µg m-3). The only potential difference is in the relative signal strengths of organic and elemental carbon, which will likely have different RIE values (Onasch et al., 2012). This is explored in more detail in section 4.2 where the relative contributions of the different factors to the HR-SP-AMS signal and the BC mass fraction is explained.

5) Re: Referee 1 comment 4: Please include the time resolution of the measurements (e.g., 10 s, 1 min, etc.). Additionally, from my understanding, the instrument was operated in different configurations (V mode, W mode, extended m/z range V mode as well as with and without the catalytic stripper). It would be useful to explicitly state the time coverage of the measurements (for example every 3rd minute for half hour of every hour).

I have included this in Instrumentation section from lines 311-320.

6) Re: Referee 1 comment 5: Please include this in the main manuscript.

Thank you so much for your comment. The reason for 1000 m/z range of this PMF analysis is that above m/z 1000, the signals were too small.

7) Re: Referee 2 comment 7: Please include the both the mass spectra (Fig. 4) and the concentration time series (Fig. 3) in the same plot. Line 222 states that Fig. 4 shows both of these, however, only the mass spectra are shown.

Figures updated as follows:

[Figure]

[Figure]

[Figure]

Figure 3. PMF solution (a) Five factors source profile (BC and MO-OOA, BBOA, Domestic burning OA, Hydrocarbon-Like OA and HOA+Fullerene) (b). The time series of non-bonfire and bonfire night factors obtained.

8) Re: Referee 2 minor comments: Please define eBCwb, eBCtr when they are first used (line 206).

Applied changes. eBCwb (Black carbon emissions from wood-burning) and eBCtr (eBCtr (Black Carbon emission from traffic).

9) Lines 22-24 "Positive matrix ....able to do." This sentence is very difficult to read. Please rewrite to improve clarity.

Positive matrix factorisation was applied to positively discriminate between different wood-burning and bonfire sources for the first time, which no existing black carbon source apportionment technique is currently able to do.

10) Lines 66- 69: "In this study ... metal nanoparticles." This should be in the experimental section, not the introduction.

Done added in experimental section.

11) Sects 2.1 and 2.2: I suggest renaming to better reflect the contents of the section. 2.1 is focused on both the site and the supporting measurements, while 2.2 is only focused on the SP-AMS rather than on all the instruments.

Renamed.

12) Sect 2.2: The text of this section needs to be revised to more clearly describe the instrument as there are several confusing/incorrect statements. For instance (not the only case), line 118 "…the new vaporiser is designed, to detect the vaporized species through electron ionization for the generation of the chemical ion…" is problematic as it implies that the vaporizer detects the species and the term chemical ion is poorly defined.

Revised.

13) Line 131: I don't understand why the term isotopes is in parentheses or why it is included in this section. Please explain or revise. The fragments should be identified as ions.

Revised and corrected.

14) Line 140: All the signal is integrated in UMR, not "can." Also "robust" in what way"

Corrected.

15) Lines 151-153: Please spend a few sentences describing the Corbin et al approach and how this approach is different. The short comment by Corbin may be of help here.

Revised text as follows:

As with all PMF analysis, error estimates have to be provided but because of the lower signals and the combination of different data retrieval method used for the fullerene signals (UMR rather than HR), greater emphasis had to be placed on these signals. Corbin et al. (2015) presented a very detailed error model for HR data

employing a Monte Carlo method to explore multiple sources of error. But because UMR was used in this instance, we were unable to apply this method, so we took an empirical approach. This was done by applying an additional 'model error' to the error matrix, i.e., an error term proportional to the signal intensity in addition to its square root, as per the standard AMS error model (Ulbrich et al., 2009; Comero et al, 2009). The model error value was increased from 0 to 0.10 to down weight the larger signals and place more of an emphasis on the fullerene signals. According to Corbin et al., 2015, the peak width 'w' is predicted during the peak fit integration from an empirical fit to the data. This 'w' prediction has a linked proportional uncertainty $\sigma w = w$. In that data set, $\sigma w = w$ was 2.5%, which was independently treated as 2 or 3% uncertainty in the isolated peaks heights, so these two can be combined in quadrature. And for the isolated peaks the value for the total percentage uncertainty is about 5% which is conceptually equivalent to 0.05 model error. This is comparable to the 0.1 model error u $\sigma w = w$ sed here. Along with placing greater emphasis on the smaller fullerene signals, the application of this model error also increased the number of "weak" variables, defined as having SNR below 2 (Paatero and Hopke, 2003; Ulbrich et al., 2009), which were down weighted by a factor of two. No variables were "bad" in the sense of having SNR < 0.2 (Paatero and Hopke, 2003).

16) Line 156: Please describe more completely how this reflects that the "data and uncertainties have a lognormal distribution." I did not find this information in the supplement.

Revised text in the main manuscript;

17) Figure 1: Why is Sr included in this time-series? It is not referenced in the text.

I have added the details about Sr peak in the text.

"During the periodic stagnation weather phenomenon, the very high concentrations of BC and Sr was also observed with the signal of 3400 s-1 and 53 s-1 respectively, during the bonfire event at 10:20 pm and 9:50 pm, as compared to (BC concentrations of 100 s-1 – 500 s-1 before Bonfire and 250 s-1 - 300 s-1 after bonfire night) and (Sr concentrations of 5 s-1 and 1 s-1 before and after bonfire night)".

18) Line 199: From the figure (2b) it appears that rBC starts increasing first. Do you mean to say that the peak signals are at different times? Please clarify.

The peak signals for rBC and organic aerosols were at different times such as the concentrations of rBC started increasing first at 07:54 pm followed by organic aerosol concentrations increasing at 8:30 pm to 9:00 pm (highest).

19) Figure 2: Please standardize the x-axes on the panels to the same range and format (date or date +time). Figure 2b: Please use standard AMS colors and remove the negative charge from the legends.

Done.

[Figure]

20) Line 226: Please describe what you mean by "mixing" so readers who are not familiar with PMF analysis more clearly understand the results.

In that case, only two unambiguously bonfire night sources of BC were identified, with a degree of 'mixing' between the bonfire night factor and traffic noted in the HOA factor, which means the factors share similar common features in their time series or profiles.

21) For the PMF results without fullerenes included, were the same optimization steps taken as for the PMF performed on the data with the fullerenes (lines 232-236)? Was a 6 factor solution explored and did that improve the results?

Firstly, the factorisation was performed without the inclusion of fullerene signals in the data matrix by choosing the f-peak between -2.0 and +2.0 with fpeak interval of 0.2 and model error of 0.10.

Yes, the six-factor solution was also explored and explained in supplementary materials. The 6-factor solution has two 'split' factors representing the same emissions. These are factor 2 and factor 4 in figure S7b and represent domestic wood burning sources because their peaks were evident before and after the bonfire night event (fig. S7a, S7b).

22) Figure 4: Please fix the y-axes so that the tick marks do not overlap and that the values can be clearly read. Please increase the resolution of the legend and replace "gt1" with ">1." Please color HO and Cx differently. In the legend, point out that the magnitude of the fullerene signal varies widely between the factors.

 Done. Please see below;

[Figure]

23) Line 345: I think you mean Figure 6 here, not 7.

Corrected.

24) Line 381-382: This sentence confuses me as it seems to imply that this work and Reyes et al suggest that fireworks are a source of soot. Perhaps you mean "consistent with the results reported here and in Reyes at all that fireworks are not soot sources."

Corrected, please see below:

The firework tracer Sr has shown some correlation with pPON and BBOA, but their peaks occurred at slightly different times. So, in spite, of the high correlation, this implies that they are not identical, and 'Sr' is behaving as a separate factor when subject to PMF analysis. It could be that if the firework display occurred at the beginning of the bonfire event their emissions maybe coincident with the pyrolysis emissions on bonfire begin to be lit, as distinct from the smouldering emissions later (Haslett et al., 2017). But without specific knowledge of the timings of the events that contribute to these emissions, it is difficult to reach firm conclusions. This, coupled with the fact that Sr could not be associated with any of the factors in this study, would be consistent with Reyes et al. (2018) in concluding that fireworks are not a significant source of the material observed.

25) Sect. 4.2: I think your figure numbering in the text versus the figures themselves is off. Figure 8 in the text seems to be what is actually labeled figure 7 and likewise for 8/9.

Corrected.

26) Supplement: Please use the HR coloring on the mass spectra as was done in the main text.

Corrected.

References

Onasch, T. B., Fortner, E. C., Trimborn, A. M., Lambe, A. T., Tiwari, A. J., Marr, L. C., Corbin, J. C., Mensah, A. A., Williams, L. R., Davidovits, P., and Worsnop, D. R.: Investigations of SP-AMS Carbon Ion Distributions as a Function of Refractory Black Carbon Particle Type, Aerosol Sci. Technol., 49, 409–422, https://doi.org/10.1080/02786826.2015.1039959, 2015.

Rivellini, L.-H., Adam, M. G., Kasthuriarachchi, N., and Lee, A. K. Y.: Characterization of carbonaceous aerosols in Singapore: insight from black carbon fragments and trace metal ions detected by a soot particle aerosol mass spectrometer, Atmospheric Chem. Phys., 20, 5977–5993, https://doi.org/10.5194/acp-20-5977-2020, 2020.

---

## Author Response (AR3)

**Minor Comments**

Dear Authors: Thank you for your consideration of the previous comments. I am happy to accept the paper for publication following attention to the minor comments below. Line numbers refer to the track changes version.

**1) Lines 329-333. Thank you for the addition of the text. However, the current wording is confusing to the reader as the description of the Corbin et al results begins by discussing peak width - a parameter not important for UMR. I suggest you start this section off with a sentence describing how the approaches give similar errors (along the lines of the last sentence of this addition) and then describe the Corbin et al approach.**

Additional details about Model error value modification are available in the supplementary material (S3a, S3b, S4a, S4b, S5a, S5b). While the methods of Corbin et al. (2015) cannot be directly applied here, they are in broad agreement with the values we have used. According to Corbin et al., 2015, the peak width 'w' is predicted during the peak fit integration from an empirical fit to the data. This 'w' prediction has a linked proportional uncertainty $\sigma w = w$. In that data set, $\sigma w = w$ was 2.5%, which was independently treated as 2 or 3% uncertainty in the isolated peaks heights, so these two can be combined in quadrature. And for the isolated peaks the value for the total percentage uncertainty is about 5% which is conceptually equivalent to 0.05 model error. This is comparable to the 0.1 model error $\sigma w = w$ used here. Along with placing greater emphasis on the smaller fullerene signals, the application of this model error also increased the number of "weak" variables, defined as having SNR below 2 (Paatero and Hopke, 2003; Ulbrich et al., 2009), which were down weighted by a factor of two. No variables were "bad" in the sense of having SNR < 0.2 (Paatero and Hopke, 2003).

**2) Line 245-246: "...ensures quantitatively." --> "...ensures quantitative measurements." or something along those lines.**

Corrected.

**3) Lines 506-507: "Variables from this and previous publications." Please specify which ones are from previous publications and provide references.**

Variables from this (such as rBC, BC & MO-OOA, BBOA, Domestic burning, Hydrocarbon-like OA and HOA+Fullerene) and previous publications such as HCNO, HCN, HONO from Priestley et al., (2018a) and sPON, pPON, $eBC_{tr}$, $eBC_{wb}$ and $eBC_{total}$ from Reyes et al., (2018).

**4) Figures 2 & 3: Please improve the resolution of the labels and the axes.**

Resolution of labels and axes are improved.